# CERTIFYING GRAPH NEURAL NETWORKS AGAINST LABEL AND STRUCTURE POISONING

## ABSTRACT

Robust machine learning for graph-structured data has made significant progress against test-time attacks, yet certified robustness to poisoning – where adversaries manipulate the training data – remains largely underexplored. For image data, state-of-the-art poisoning certificates rely on partitioning-and-aggregation schemes. However, we show that these methods fail when applied in the graph domain due to the inherent label and structure sparsity found in common graph datasets, making effective graph-partitioning difficult. To address this challenge, we propose a novel semi-supervised learning framework called deep Self-Training Graph Partition Aggregation (ST-GPA), which enriches each graph partition with informative pseudo-labels and synthetic edges, enabling effective certification against node-label and graph-structure poisoning under sparse conditions. Our method is architecture-agnostic, scales to large numbers of partitions, and consistently and significantly improves robustness guarantees against both label and structure poisoning across multiple benchmarks, while maintaining strong clean accuracy. Overall, our results establish a promising direction for certifiably robust learning on graph-structured data against poisoning under sparse conditions.

## 1 INTRODUCTION

Graph Neural Networks (GNNs) are highly susceptible to adversarial perturbations in their input graph applied at test or training time (Zügner et al., 2018). Subsequently developed empirical defenses are at the continual risk of being rendered ineffective by more sophisticated ways to choose adversarial perturbations (Koh et al., 2022; Mujkanovic et al., 2022). This motivates the development of robustness certificates, which provide provable guarantees about the stability of predictions under worst-case data perturbations, allowing to rigorously assess and mitigate adversarial vulnerabilities. While significant advances in providing such provable guarantees for GNNs against test-time attacks have been made (Günnemann, 2022b; Hojny et al., 2024), certifying robustness of GNNs against data poisoning, where an adversary can manipulate the graph structure (Zügner & Günnemann, 2019) or node-labels (Lingam et al., 2024) at training time, remains largely underexplored.

The most effective approaches to derive poisoning robustness guarantees in the image domain rely on partitioning the training data, learning separate (base) classifiers, and aggregating predictions (e.g., via majority vote) (Levine & Feizi, 2021). However, we demonstrate that these methods fail when directly applied to common graph learning tasks such as node classification. In particular, we find that the core challenge is *sparsity*: both labels and graph structure are often too sparse to provide effective training signals under data partitioning, leading to poor performance for label certificates and vacuous robustness guarantees for structure poisoning. This raises a critical question:

> *How can we effectively analyze and guarantee the trustworthiness of graph neural networks in the presence of structure and label poisoning, while maintaining their utility?*

In this work, we address this challenge by introducing a novel semi-supervised learning framework called deep Self-Training Graph Partition Aggregation (ST-GPA) that successfully overcomes the sparsity problem and enables effective poisoning certification of GNNs against structure and label poisoning (see Figure 1). In particular, our method enriches the subgraphs created through partitioning the original graph, with synthetic data generated using carefully designed self-training approaches. *Self-training* is a concept from semi-supervised learning (Chapelle et al., 2006) that refers to training

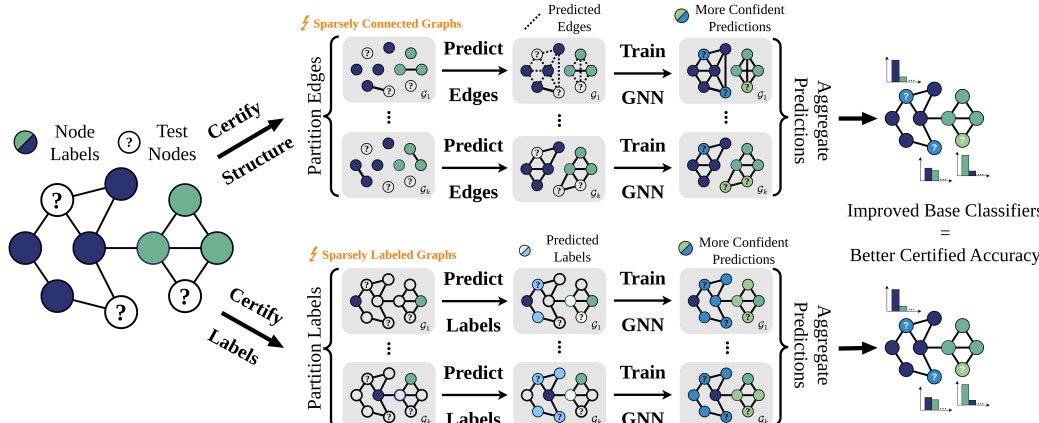

Figure 1: Deep Self-Training Graph Partition Aggregation (ST-GPA). To certify against *structure poisoning*, the graph's edges are partitioned. The resulting set of sparse graphs are enriched with informative pseudo-edges and then, an ensemble of GNNs is trained on the partitions. To certify against *label poisoning*, the graph's node labels are partitions, and the resulting sparsely labeled graphs enhanced with pseudo-labels, before training the ensemble. To *jointly certify* against both poisoning, ST-GPA partitions both the edges and labels and adds pseudo-edges and pseudo-labels.

a model on its own predictions to expand the leverageable training set, and it has been successfully used in empirical defenses such as Li et al. (2024); Lee & Park (2025). Concretely, for each sparsely labeled graph partition, our method efficiently generates synthetic (pseudo) labels, expanding the available node-label set. In a similar spirit, for each sparsely connected subgraph, our method generates synthetic edges through solving a link prediction problem, endowing each subgraph with a more informative structure. Then, by training GNNs as base classifiers on these enriched subgraphs and aggregating their outputs using a majority vote, we obtain strong certificates for label poisoning, structure poisoning, and joint label-structure poisoning. Our contributions are:

**(i)** In Section 3 we generalize the partition-and-aggregate paradigm prevalent in the image domain to derive poisoning robustness certificates to non-i.i.d structured data. Then, in Section 4, we **identify and characterize a failure mode** of certifying robustness against poisoning using this paradigm **on graph-structured data** rooted in the label and structure sparsity found in common graph datasets.

**(ii)** In Section 5 we propose deep Self-Training Graph Partition Aggregation (ST-GPA), a novel semi-supervised learning framework that augments each partition with carefully obtained pseudo-labels and/or synthetic edges, **overcoming the sparsity problem** when partitioning graph-structured data.

**(iii)** ST-GPA is the first effective certificate against structure and label poisoning for node classification with GNNs. In particular, we set a **new state-of-the-art certified poisoning robustness for GNNs** across multiple benchmarks, while being efficient with competitive clean accuracies (see Section 6).

Overall, our work highlights the importance of self-training in certifying poisoning robustness of GNNs and lays the foundation for future work against training-time attacks in the graph domain. We believe that our insights on improving partition-based certification using self-training may be of independent interest beyond graph learning.

## 2 PRELIMINARIES

We consider an attributed, undirected graph $G \subseteq \mathbb{G}$ described by a feature matrix $\boldsymbol{X} \in \mathbb{R}^{n \times m}$ of $m$-dimensional features for $n$ nodes and a set of edges $\mathcal{E} \subseteq \{\{i,j\}, i,j \in [n]\}$ with $[n] = \{1,\ldots,n\}$, where two nodes $i$ and $j$ are connected if and only if $\{i,j\} \in \mathcal{E}$, and each node belongs to one out of $C$ classes with $\mathbb{Y} = [C]$. We study semi-supervised (transductive) node classification, that is the full graph $G = (\mathcal{E}, \boldsymbol{X}, \boldsymbol{y})$ is available at training but its nodes are only partially labeled. This is modeled using a label vector $\boldsymbol{y} \in (\mathbb{Y} \cup \{-1\})^n$, where entry $y_i \in \mathbb{Y} = [C]$ indicates the label of node $i$, and $y_i = -1$ means the node $i$ is unlabeled. We model GNNs as functions $f : \mathbb{G} \to \mathbb{Y}^n$, and collect the labels in the graph in the set $\mathcal{Y} = \cup_{v \in V_{\text{labeled}}} \{(v, \boldsymbol{y}_v)\}$ and the node attributes in the set $\mathcal{X} = \cup_{v \in V} \{(v, \boldsymbol{X}_v)\}$. We write graph datasets as $D = (\mathcal{E}, \mathcal{X}, \mathcal{Y})$.

**Perturbation model.** In this work we study two types of poisoning attacks on graph-structured data: *label flipping* and *structure perturbations*. To quantify the strength of an attack, we define *attack budgets* $r_l, r_s$, which bound the number of allowed modifications to labels and structure, respectively. We model the set of possible poisoned graphs $\tilde{G}$ as a ball centered around the clean graph $G$:

$$\mathcal{B}_{r_l, r_s}(G) = \left\{ \tilde{G} = (\tilde{\mathcal{E}}, \boldsymbol{X}, \tilde{\boldsymbol{y}}) \mid \delta(\tilde{\boldsymbol{y}}, \boldsymbol{y}) \leq r_l, \Delta(\tilde{\mathcal{E}}, \mathcal{E}) \leq r_s \right\} \tag{1}$$

where $\Delta(\mathcal{X}, \mathcal{Y}) = |(\mathcal{X} \setminus \mathcal{Y}) \cup (\mathcal{Y} \setminus \mathcal{X})|$ is the symmetric difference between two sets, and $\delta(\tilde{\boldsymbol{y}}, \boldsymbol{y}) = \sum_{i=1}^{n} \mathbf{1}_{\tilde{y}_i \neq y_i}$ is the number of different entries between two vectors (i.e. Hamming distance). Equation (1) models three different perturbation models: (i) label flipping ($r_l > 0, r_s = 0$); (ii) structure poisoning ($r_l = 0, r_s > 0$); and (iii) both together ($r_l > 0, r_s > 0$).

**Deep partition aggregation.** State-of-the-art certified robustness guarantees for supervised image classification under poisoning attacks have been achieved via the partition-and-aggregate paradigm. The most prominent implementation of which is Deep Partition Aggregation (DPA) (Levine & Feizi, 2021). Here, the core idea is to partition an i.i.d training dataset $D$ into $k$ disjoint subsets and to train independent classifiers $f_D^{(i)} : \mathcal{X} \to \mathcal{Y}$ deterministically on each partition $i$. At inference time, for a given input image $x \in \mathcal{X}$, the final prediction is made via *majority vote* across all base classifiers:

$$g_D(x) = \arg \max_{c \in \mathcal{Y}} n_c(D, x), \tag{2}$$

where $n_c(D, x) = \sum_{i=1}^{k} \mathbf{1}[f_D^{(i)}(x) = c]$ counts the number of classifiers predicting class $c$ for input $x$. This setup enables a formal robustness guarantee for the aggregated prediction under poisoning:

**Theorem 1** (Levine & Feizi (2021)). *Given a clean dataset $D$, the majority-vote prediction remains unchanged, i.e. $g_D(x) = g_{\tilde{D}}(x)$, for any perturbed dataset $\tilde{D} \in \mathcal{B}_r(D)$ bounded by the attack budget $r$, as long as*

$$r \leq \left\lfloor \frac{n_c(D, x) - \max_{c' \neq c} (n_{c'}(D, x) + \mathbf{1}_{c' < c})}{2} \right\rfloor, \tag{3}$$

*where $c = g_D(x)$ is the predicted class on the clean dataset.*

## 3 GENERALIZED DEEP PARTITION AGGREGATION FOR NON-I.I.D. DATA

Even though DPA has been formulated for image data, the general paradigm to derive poisoning guarantees can be readily generalized to non-i.i.d. structured data. The main idea is based on recognizing that deriving a guarantee like Theorem 1 does not depend on the i.i.d. nature of the dataset, but rather on a partitioning scheme $h$ where the potentially poisoned objects are partitioned into subsets *independent* of one another. In particular, it can be formulated w.r.t. a general set of objects $\mathcal{O}$ that may be poisoned, where $o_1 \in \mathcal{O}$ may not be independent of another $o_2 \in \mathcal{O}$ (e.g., in a graph, one node may not be independently sampled from another node), as long as the partition $o_1$ is grouped into, is not affected by the value of any other element in $\mathcal{O}$, and thus, poisoning any object can only affect exactly one partition.

Concretely, to formulate a split-and-majority voting certificate for general potentially structured data, assume a dataset $D = (\mathcal{T}, \mathcal{O})$ consists of a set of objects $\mathcal{T}$ that are known to be clean and a set of objects $\mathcal{O}$ that may be poisoned. For example, for i.i.d. image data $\mathcal{O} = \{x_i\}_{i=1}^{n}$ and $\mathcal{T} = \varnothing$. Then, $k$ partitions $\mathcal{P}_i$ with $i \in [k]$ are created as follows:

$$\mathcal{P}_i := (\mathcal{T}, \{o \in \mathcal{O} \mid h(\mathcal{T}, o) = i \pmod{k}\}) \tag{4}$$

where $h$ is a deterministic (hash) function that takes $\mathcal{T}$ and one $o \in \mathcal{O}$ as input and outputs a number in $\mathbb{N}$ representing the partition index, into which the given object $o$ should be grouped. Then, one independently trains base classifiers $f_D^{(i)} : \mathbb{X} \to \mathbb{Y}$ on each partition $\mathcal{P}_i$, where $\mathbb{X}$ represents a general data domain. Without loss of generality, we assume $f_D^{(i)}$ to output a scalar class prediction, i.e., $\mathbb{Y} \subseteq \mathbb{N}$. For example, for node classification, the data domain can be defined as $\mathbb{X} = (\mathbb{G}, \mathbb{N}_0)$ where $\mathbb{G}$ is the set of possible graphs and the second element in the tuple refers to the index of a node, for which a class prediction is sought.[1] Now, given an input $x \in \mathbb{X}$, the final prediction is made as in DPA for images via a majority vote across all base classifiers (Equation (2)), which we denote $g_D(x)$.

---

[1] If $f_D^{(i)}$ outputs a vector for several indexable objects (e.g., nodes in a graph), this can be equivalently represented as a scalar prediction for each indexable object, where the index of the object for which the prediction is sought for, is part of $\mathbb{X}$.

Lastly, for general data domains, the set difference $\Delta(D, \tilde{D})$ may not always be an appropriate distance measure, and different choices of the hash function $h$ may affect how many partitions $d_h \in \mathbb{N}$ are affected when poisoning an object $o \in \mathcal{O}$. Thus, assume a general distance function $d(D, \tilde{D})$ between the clean dataset $D$ and the perturbed dataset $\tilde{D}$. Further, assume that a perturbation of size $d(D, \tilde{D})$ leads to at most $p$ changed partitions given $h$. Then, the scalar $d_h$ links the perturbation size $d(D, \tilde{D})$ to the upper bound $p$ on the number of changed partitions as follows: $d_h \cdot d(D, \tilde{D}) \geq p$. Exemplary, $d_h > 1$ if one object is partitioned (duplicated) into multiple partitions as done by Wang et al. (2022). Now, we can state the following general theorem that follows the proof strategy outlined by Levine & Feizi (2021) and we refer to Appendix C.1 for a formal proof:

**Theorem 2** (Generalized DPA). *Given a clean, possible non-i.i.d and structured dataset $D$, and a poisoned dataset $\tilde{D}$, the majority-vote classifier prediction remains unchanged, i.e., $g_D(x) = g_{\tilde{D}}(x)$, as long as $d_h d(D, \tilde{D}) \leq \lfloor (n_c(D, x) - \max_{c' \neq c} (n_{c'}(D, x) + \mathbb{1}_{c' < c}))/2 \rfloor = r_m(D, x)$, where $c = g_D(x)$ is the predicted class on the clean dataset.*

We refer to the right-hand side of the condition in Theorem 2 as the *robust margin* $r_m(D, x)$ of a sample $x$ given a dataset $D$. To conclude, we need to find a (hash) function $h$, a scalar $d_h$, and a distance measure $d(D, \tilde{D})$ to get a robustness guarantee for an ensemble classifier with Theorem 2.

**Generalized DPA for Graphs.** We define a general hash function in Equation (5) that allows to partition a graph based on its node features $\mathcal{X}$. This hash function can be used for all three perturbation models captured by Equation (1), as we detail below.

$$h(\mathcal{X}, o) = \begin{cases} h(X_i \| X_j) + h(X_j \| X_i) & \text{if } o = \{i, j\} \in \mathcal{E} \\ h(X_v) & \text{if } o = (v, \boldsymbol{y}_v) \in \mathcal{Y} \end{cases} \quad (5)$$

The hash function $h(\mathcal{X}, o)$ determines the partition index for an edge, by taking the features of the incident nodes and concatenating them. We add both orderings to make the partitioning process invariant to the actual node order. When partitioning labels, $h(\mathcal{X}, o)$ takes the corresponding node's features as input. The $h$ in Equation (5) can be any hash function, but we choose the MD5 hash in this work to make it simple to take in strings and convert the output to non-negative integers. As each edge or label is in exactly one partition, $d_h = 1$.

Finally, we choose three distance metrics for the three different threat models. For *label poisoning*, we take the Hamming distance $\delta(\boldsymbol{y}, \tilde{\boldsymbol{y}})$ between label vectors. For *graph structure poisoning*, we use the symmetric set difference $\Delta(\mathcal{E}, \tilde{\mathcal{E}})$ between edge sets. Note that inserting or deleting an edge in the graph leads to $\Delta(\mathcal{E}, \tilde{\mathcal{E}}) = 1$ and the modification of an edge to $\Delta(\mathcal{E}, \tilde{\mathcal{E}}) = 2$. Similarly, the insertion or deletion of an edge leads to exactly one affected partition based on $h$, but a modification of an edge affects two. For a *combination of label and structure poisoning*, we add up the two distances $d((\mathcal{E}, \boldsymbol{y}), (\tilde{\mathcal{E}}, \tilde{\boldsymbol{y}})) = \delta(\boldsymbol{y}, \tilde{\boldsymbol{y}}) + \Delta(\mathcal{E}, \tilde{\mathcal{E}})$. This is indeed a distance metric and the proof can be found in Appendix C.2. Given the graph dataset $D$ from Section 2, plugging in the different distance measures along with our $h$ from Equation (5) and $d_h = 1$ into Theorem 2, we get the following conditions for our certificates to hold:

i. $\tilde{G} \in \mathcal{B}_{r_l > 0, r_s = 0}(G): \quad \delta(\boldsymbol{y}, \tilde{\boldsymbol{y}}) \leq r_m(D, x) \qquad$ (label-flipping certificate) $\qquad$ (6)

ii. $\tilde{G} \in \mathcal{B}_{r_l = 0, r_s > 0}(G): \quad \Delta(\mathcal{E}, \tilde{\mathcal{E}}) \leq r_m(D, x) \qquad$ (structure-poisoning certificate) $\quad$ (7)

iii. $\tilde{G} \in \mathcal{B}_{r_l > 0, r_s > 0}(G): \quad d((\mathcal{E}, \boldsymbol{y}), (\tilde{\mathcal{E}}, \tilde{\boldsymbol{y}})) \leq r_m(D, x) \qquad$ (certifying both) $\qquad$ (8)

where $r_m(D, x)$ is the robust margin introduced in Theorem 2. In the context of transductive node classification, $x$ refers to a node that we seek to classify in the graph $D$. In general, $x$ could refer to a node in an arbitrary graph different to $D$.

# 4 LIMITATIONS OF THE SIMPLE PARTITIONING IN THE GRAPH DOMAIN

We first expose that the partitioning scheme used by DPA does not work well on graph datasets due to their inherent label and structure sparsity. In the context of image classification, obtaining provable poisoning robustness through partitioning yields significant results (Levine & Feizi, 2021). For example, using 1,000 partitions on CIFAR-10 achieves a certified accuracy of 50% against 392 label

flips. The key factor in deriving strong robustness guarantees through partitioning is the number of partitions $k$. Increasing the number of partitions increases the possible perturbation budget. Notably, the maximum number of tolerated perturbations before the certified ratio drops to 0% is $\lfloor \frac{k}{2} \rfloor$. As a result, the number of partitions $k$ should be as large as possible to produce good certified radii. In image datasets, the primary constraint on increasing $k$ is the size of the training dataset. With 50,000 training images in CIFAR-10, it is feasible to use $k = 1,000$ partitions while maintaining acceptable performance of the base classifiers $f_D^{(i)}$.

However, this approach does not translate straightforwardly into the graph domain. In particular, common graph datasets typically contain significantly fewer labeled training samples (nodes) compared to image datasets (see Table 1 in Appendix A). For label-flipping attacks, applying the partitioning scheme naively to node labels results in partitions with very few labeled nodes – often fewer than the number of classes. For example, as illustrated in Figure 2, when using 40% of nodes for training on Cora-ML, having $k = 281$ partitions yields an average of only 4 labeled nodes per partition. This restricts the certification bound to 140 label flips, and base classifiers suffer from significantly reduced performance, as each model only has access to on average 4 labels during training. Even worse, Figure 4(a) shows that choosing $k = 80$ partitions already leads to a clean accuracy of only close to 40% of an ensemble of GCNs (Kipf & Welling, 2017) on Cora-ML, whereas a GCN trained without partitioning achieves ∼78.77% accuracy.

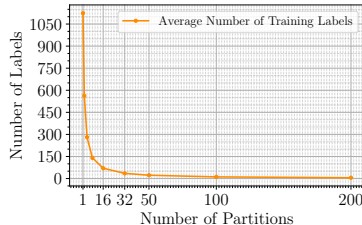

Figure 2: Label sparsity (Cora-ML)

Similarly, for structure partitioning the graph structure rapidly deteriorates as edges are divided among partitions, rendering the partitions ineffective for GNNs. This is illustrated in Figure 3, where we plot the mean of average node degrees across partitions. When the number of partitions roughly exceeds $k > 50$, the connectivity in the individual partitions' subgraph becomes virtually nonexistent. Since GNNs rely on various forms of message passing to aggregate information from graph neighborhoods, partitions with sparse or disconnected edges severely impair their ability to learn meaningful representations. Under such conditions, the base classifiers' performance deteriorate to that of a multilayer perceptron (MLP), which, however, is already inherently robust to structure perturbations since it does not utilize potentially poisoned edge information.

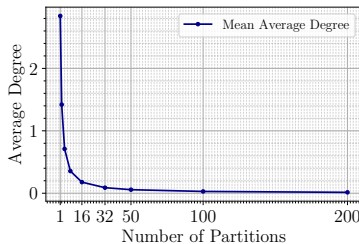

Figure 3: Graph structure sparsity (on Cora-ML)

Overall, partitioning-based robustness certificates cannot be naively applied in the graph domain due to the inherent label and structure sparsity, motivating the need for more sophisticated methods.

## 5 DEEP SELF-TRAINING GRAPH PARTITION AGGREGATION

To address these sparsity challenges, we propose various semi-supervised learning methods to enhance the performance of the weak classifiers by selecting either pseudo-labels or edges, or both, on each partition's limited training data. In this way, we obtain strong certificates on graph datasets.

### 5.1 SEMI-SUPERVISED LEARNING FOR LABEL GENERATION

To maximize the utilization of the limited available labels within each partition, we employ two complementary pseudo-label generation methods: (i) co-training (CT), and (ii) self-training (ST). *First*, the co-training method leverages the graph structure to propagate existing labels to neighboring nodes. While the previous work (ParWalks, Wu et al. (2012)) also generates pseudo-labels for the training of a GNN as proposed by Li et al. (2018), its high computational complexity renders this approach impractical for larger graph datasets. Instead, we propose the use of label propagation (LP) (Zhu & Ghahramani, 2003) to significantly speed up the co-training process. The key advantage of LP is that it does not require computing the inverse of a Laplacian matrix inherent to ParWalks, and can be efficiently applied to large graph datasets. *Second*, for self-training we propose training a GNN on the existing labeled data and selecting the most confident predictions, as determined by their softmax scores, to serve as pseudo-labels for subsequent training iterations. Notably, both methods

**Algorithm 1** ST-GPA against label flipping

**Require:** Graph dataset $D = (\mathcal{E}, \mathcal{X}, \mathcal{Y})$, selectiveness $t$, co-training method $\text{CT}(\mathcal{E}, \mathcal{X}, \boldsymbol{y}, t)$, self-training method $\text{ST}(\mathcal{E}, \mathcal{X}, \boldsymbol{y}, t)$, training order $o_t = \{\text{CT}, \text{ST}\}^m$, hash function $h$

**Ensure:** A robust ensemble classifier $g$

1: split labels into partitions
   $\boldsymbol{y}_i = \{y | y \in \mathcal{Y}, h(\mathcal{X}, y) \equiv i \pmod{n}\}$
2: **for** each partition $i$ **do**
3:   **for** each operation $op$ in given order $o_t$ **do**
4:     $\hat{\boldsymbol{y}} = op(\varepsilon_i, \mathcal{X}, \boldsymbol{y}_i, t)$
5:     $\boldsymbol{y}_i = \boldsymbol{y}_i \cup \hat{\boldsymbol{y}}$
6:   **end for**
7:   train $f_i$ on $\hat{D} = \{\mathcal{E}, \mathcal{X}, \boldsymbol{y}_i\}$
8: **end for**
9: count base classifier predictions $n_c(v)$
10: **return** $g(v) = \arg\max_{c \in [C]} n_c(v)$

**Algorithm 2** ST-GPA against structure pert.

**Require:** Graph dataset $D = (\mathcal{E}, \mathcal{X}, \mathcal{Y})$, selectiveness $\varepsilon$, link prediction method $\text{LinkPred}(e_i, \mathcal{X}, \mathcal{Y}, \varepsilon)$, hash function $h$

**Ensure:** A robust ensemble classifier $g$

1: split edges into partitions
   $e_i = \{e | e \in \mathcal{E}, h(\mathcal{X}, e) \equiv i \pmod{n}\}$
2: **for** each partition $i$ **do**
3:   $\hat{e}_i = \text{LinkPred}(e_i, \mathcal{X}, \mathcal{Y}, \varepsilon)$
4:   $e_i = e_i \cup \hat{e}_i$
5:   train $f_i$ on $\hat{D} = \{e_i, \mathcal{X}, \mathcal{Y}\}$
6: **end for**
7: count base classifier predictions $n_c(v)$
8: **return** $g(v) = \arg\max_{c \in [C]} n_c(v)$

(co- and self-training) can be applied consecutively as we demonstrate in Section 6, where we observe that applying first co-training and then self-training works best (Figure 5(a)).

In LP we propagate a score matrix $\boldsymbol{S} \in \mathbb{R}^{n \times c}$ in which $S_{i,j}$ represents the likelihood of node $i$ belonging to class $j$. $\boldsymbol{S}$ is initialized to be the one-hot encoded label matrix $\boldsymbol{Y} \in \mathbb{R}^{n \times c}$ as in Equation (9) (where an unlabeled node has a row of zeros in $\boldsymbol{Y}$), and propagated with the normalized adjacency matrix $\tilde{\boldsymbol{A}} = \boldsymbol{D}^{-\frac{1}{2}} \boldsymbol{A} \boldsymbol{D}^{-\frac{1}{2}}$, with a probability of $1 - \alpha$ to randomly teleport to a labeled node, as shown in Equation (10), which is iterated until convergence or a cutoff iteration.

$$\boldsymbol{S}^{(0)} = \boldsymbol{Y} \tag{9}$$
$$\boldsymbol{S}^{(i+1)} = \alpha \tilde{\boldsymbol{A}} \boldsymbol{S}^{(i)} + (1 - \alpha) \boldsymbol{Y} \tag{10}$$

Both co-training and self-training result in a score matrix $\boldsymbol{S} \in \mathbb{R}^{n \times c}$, which is used for selecting pseudo-labels. We introduce hyperparameter $t$ representing the number of pseudo-labels with the highest scores added *per class* via either method to control the selectiveness. A lower $t$ adds less but often higher-quality pseudo-labels, while higher $t$ adds more but less-confident ones. Using these methods, we present the full certification pipeline against label flipping in Algorithm 1.

### 5.2 SEMI-SUPERVISED LEARNING FOR EDGE GENERATION

Motivated by the concept of self-training in graph machine learning, we extend semi-supervised learning to edge prediction by generating pseudo-edges instead of pseudo-labels. Guided by the homophily assumption, we generate pseudo-edges that connect node pairs likely belonging to the same class based on a model trained on the current graph. Here, modern link prediction methods (Zhang & Chen, 2018; Kazi et al., 2023) struggle as the edges in every partition are extremely sparse (see Section 4). Instead, we try to capture the overall graph structure and dilute the effect of possible inter-class edges by adding in a denser graph than the original one. Within each partition, we first train a GNN to obtain initial node predictions and confidence scores, typically represented by the softmax outputs of the final GNN layer. We then iteratively add edges between node pairs with the highest sum of confidence of being in the same class. The number of edges added is controlled by a hyperparameter $\varepsilon$, which specifies the multiplier of the number of added edges within each partition relative to the original graph. Using this expanded edge set, a second GNN is trained to produce final predictions used for certification. Our certification pipeline against structure perturbation is outlined in Algorithm 2. On first sight, a downside of our link prediction scheme is that we sample a denser graph for each partition than the original one. If this is implemented naively, it will have roughly $O(kn^2)$ time and memory complexity, which does not scale well with the graph size and the number of partitions. To address this, we introduce an efficient algorithm in Appendix D.1, which has a near-linear complexity based on managing a global edge-candidate heap.

With the proposed methods addressing either label flipping or structure perturbations, a scheme for certifying against *both types of attacks* becomes possible. We partition both the poisoned labels and

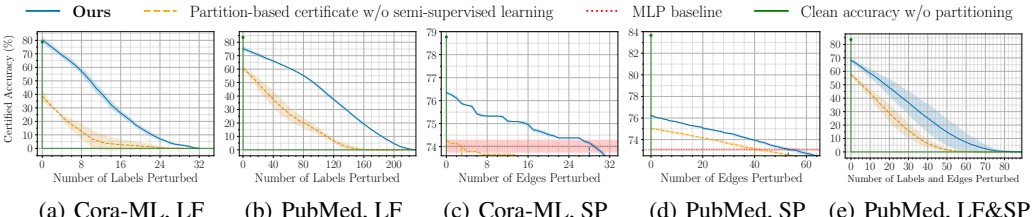

| (a) Cora-ML, LF | (b) PubMed, LF | (c) Cora-ML, SP | (d) PubMed, SP | (e) PubMed, LF&SP |

Figure 4: The effectiveness of our proposed method demonstrated by the increase in certified accuracy, compared to the vanilla partition-based approach. (a) and (b) are certified against label flipping (LF) with $k = 80$ and $k = 500$ partitions repectively; (c) and (d) are certified against structure perturbations (SP) with $k = 800$ partitions; (e) is certified against both (LF&SP) with $k = 250$ partitions. We showcase the stark improvement of robustness both on Cora-ML and PubMed, demonstrating that our method works for both smaller and larger graphs. The red lines in (c) and (d) represents the performance of an infinitely robust MLP, serving as a trivial baseline for structure perturbation. The green dots are the non-robust clean accuracies of the same model trained without any partitions.

edges into partitions, perform semi-supervised edge and label generation iteratively, and obtain the final base classifier by training on the extended edges and labels, before we take the majority vote and compute the certificates.[2] We provide pseudocode for this joint pipeline in Appendix D.2.

## 6 EXPERIMENTAL EVALUATION

In this section, we investigate the robustness guarantees derived by deep self-training graph partition aggregation and showcase the improvement of ST-GPA compared to vanilla partition aggregation on graphs. We provide code to reproduce our results in the reproducibility statement.

**Experimental details.** We demonstrate results for transductive node classification on four datasets: Cora-ML (Bojchevski & Günnemann, 2018), and the three Planetoid datasets CiteSeer, Cora, and PubMed (Yang et al., 2016); and for three GNNs: Graph Convolutional Networks (GCN) (Kipf & Welling, 2017), Graph Attention Networks (GAT) (Veličković et al., 2018), and APPNP (Gasteiger et al., 2018). To train a robust classifier, we partition the datasets as described in Section 5. Each round of semi-supervised learning using ST-GPA adds pseudo-labels or edges to the training sets of the individual partitions, while keeping the partitions isolated. We train an ensemble classifier after each round of pseudo-label or pseudo-edge generation to investigate the effect of the individual semi-supervised learning steps. Results are reported using *certified accuracy*, which is the percentage of test nodes whose predictions are provably correct, as a function of the perturbation size defined in Equation (1). In all figures, the colored areas represent the standard deviation over 3 deterministically chosen seeds. We represent a baseline ensemble classifier trained on partitions without any semi-supervised learning as dashed lines. It is important to note that unlike as in label flipping, Multi-Layer Perceptrons (MLPs) exhibit infinite robustness against structure perturbations, since they do not utilize edge information during training. Consequently, any model with certified accuracy against structure perturbations below that of an MLP is trivial. Thus, we include an MLP as baseline as a red-dotted line in the structure perturbation plots. In these plots, we discard the trivial part of the curve below this baseline. We provide further details on our experiment setup in Appendix A.

**ST-GPA yields strong certified robustness.** Figure 4 demonstrates the stark improvement of certified accuracies by our proposed certification method ST-GPA, against label flipping (Figures 4(a) and 4(b)), structure perturbation (Figures 4(c) and 4(d)), and both label and structure perturbations (Figure 4(e)). If the partitioning is applied without our proposed semi-supervised learning strategy, we only get marginal certified accuracy curves (orange dashed lines), due to the sparsity of labels or edges as discussed in Section 4. In the structure case on Cora-ML (Figure 4(c)), the clean accuracy of an ensemble GCN is even worse than an infinitely robust MLP, rendering the naive approach ineffective. In contrast, with our method we restore the ensemble classifier's clean accuracy to a higher level compared to an ensemble without semi-supervised learning, typically around 70% to 80%, and this also allows the certified accuracy curves to drop down slower, meaning higher certified accuracy against the same number of perturbations. We included the non-robust clean accuracy of a GCN

---

[2]We do edge generation first, since we find co-training relies on meaningful graph structure.

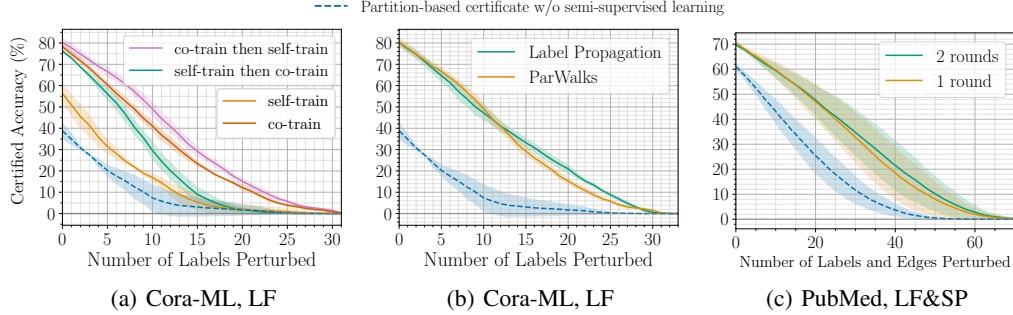

(a) Cora-ML, LF      (b) Cora-ML, LF      (c) PubMed, LF&SP

Figure 5: (a) Different orders of co-training and self-training against label flipping on Cora-ML with $k = 80$ and label propagation as co-training method; (b) Label propagation provides similar performance in co-training compared to ParWalks (Wu et al., 2012) while being scalable to larger graphs, demonstrated on Cora-ML with 80 partitions; (c) Our method provides significant improvements on PubMed with 200 partitions with link prediction, co-training then self-training, while stacking more rounds of semi-supervised learning in said order yields further but marginal improvement.

trained without any partitioning as a reference. Table 2 in Appendix A reports clean accuracies on other datasets with other models.

In our certificate against label flipping, we demonstrate the effect by the order of co-training and self-training in Figure 5(a). We find that first co-training and then self-training generally works best. This is because doing self-training first means training GNNs on very sparsely labeled partitions. With $k = 80$ partitions, each partition contains on average only about 10 labels on smaller datasets. In contrast, co-training with label propagation does not require a preliminary training step and effectively leverages the graph structure despite the extremely low label rate, producing high-quality pseudo-labels that can be further improved by subsequent self-training rounds. This effect is prevalent in all datasets as we show in Figure 8 in Appendix B.2. Consequently, we adopt this order for experiments with all other attack models involving label flipping.

In our certificate against both label flipping and structure perturbations, we perform link prediction, then co-training with label propagation, and finally self-training on labels. We perform several rounds

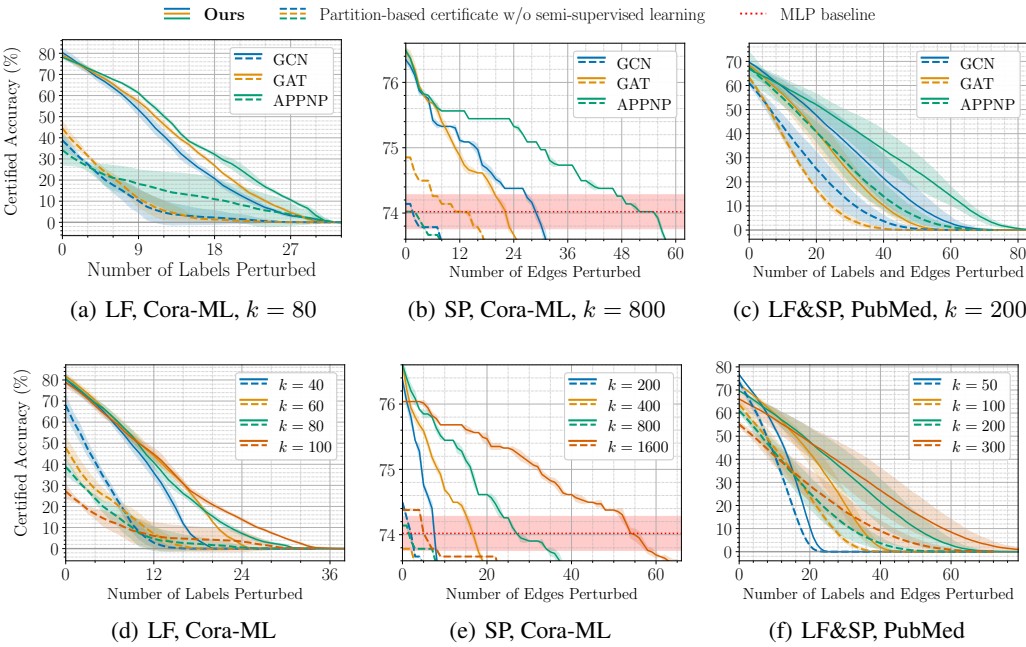

(a) LF, Cora-ML, $k = 80$    (b) SP, Cora-ML, $k = 800$    (c) LF&SP, PubMed, $k = 200$

(d) LF, Cora-ML      (e) SP, Cora-ML      (f) LF&SP, PubMed

Figure 6: In (a) to (c) our method improves the vanilla partition-based certificate regardless of the GNN type; (d) to (f) demonstrate for GCNs that our method scales very well for larger number of partitions $k$, which provides better certificates.

of edge and label generation in this order because label propagation relies on the meaningful graph structure that link prediction generates, and self-training generally works better after co-training, as we find out in Figure 5(a). In Figure 5(c) we show that our method introduces significant improvements in the first round, and stacking more rounds of edge and label generation leads to further, yet marginal, improvements. In Figure 9 in Appendix B.2 we show that our method introduces clean accuracy improvements even for the smaller graph datasets, although only with very few partitions.

**Co-training with label propagation is similarly performant as ParWalks yet scalable.** Since Li et al. (2018) originally proposed ParWalks in combination with self-training to address low label rates, we compare ParWalks with label propagation (Zhu & Ghahramani, 2003). As shown in Figure 5(b), the performance advantage of ParWalks over LP is negligible. This is consistent across different datasets as shown in Figure 10 in Appendix B.2. Given that label propagation with random teleportation does not require computing the inverse of the Laplacian, we adopt LP as primary method.

**Our method works with any GNN.** The first row in Figure 6 shows the performance of GCN, GAT, and APPNP with and without our proposed approach. The results clearly indicate that similar performance improvements and trends hold across all models, with clean accuracy boosted to around 80% for label, 76% for structure, and about 67% for both. We highlight this feature because the vanilla partition-based certificates do not assume anything on the classifier itself, and with better GNNs our method is still compatible for producing even better certificates.

**Our method scales well with partitions.** The second row in Figure 6 shows the effect of varying the number of partitions $k$. The figures clearly demonstrate the effectiveness and scalability of the proposed semi-supervised learning methods: as $k$ increases, which is necessary to derive stronger guarantees, the baseline performance rapidly declines due to the sparsity of labels and structure per partition. However, our experiments indicate that link prediction, co-training and self-training are required for non-trivial robustness guarantees. This is supported by more results in Figure 11 in Appendix B.2. We note that dividing labels to more than 100 partitions becomes impractical given the training size of 30% in the case of Cora-ML, as partitioning would result in some partitions containing no labeled nodes, making training on those partitions infeasible.

Further experiments regarding how we choose the selectiveness hyperparameter $t$ and $\varepsilon$, and label propagation teleportation parameter $\alpha$ are included in Appendix B.3.

## 7 DISCUSSION ON LIMITATIONS

We've shown that our semi-supervised training scheme is essential for meaningful robustness guarantees in the graph domain with partition-based methods, and it's powerful and scalable as it applies to any GNN and any number of partitions. In this section, we address the limitations in this scheme and show that it hints promising research directions.

**Application to heterophilic graphs.** The core scheme introduced by this paper are semi-supervised learning methods on partitioned labels or edges, namely co-training with label propagation, self-training, and link prediction. These methods are designed to leverage the homophily assumption, i.e. nodes with the similar nodes are more likely to be connected. This dependence is exposed with our evaluation of structure certificates on Wiki-CS(Mernyei & Cangea, 2020), where the graph neighborhoods are way less homogeneous. Consequently, our link prediction guided by homophily has little effect of capturing the real graph structure, yielding minimal improvements compared to a trivial partitioning scheme. In general, heterophilic graphs are an emerging research interest in the graph learning domain and has a wide range of real-world applications. Applying our method to them requires heterophilic adaptations of the semi-supervised learning methods, which are individual research directions on their own. Thus, certified accuracies on heterophilic graphs is beyond the scope of this paper. Nonetheless, the introduced partitioning and self-training scheme should still provide strong robustness guarantees when adapted to heterophilic graphs.

**Link prediction can be expensive.** In the link prediction against structure perturbations, we tried to recreate the graph structure isolating the influence of poisoned edges in each partition. Unlike real-world graphs, the graphs we created by linking intra-class edges are, by design, denser. This is inherently due to the low signal-to-noise ratio of the link prediction process – the pseudo-edges generated are, by a small chance, false. To counter this, we typically add in more edges than the original graph has, and the idea is to capture the overall graph structure and dilute the effect of

inter-class edges. Consequently, the graph we train our classifiers on is semi-dense. This result in a higher overhead for the training process, both in computation and memory. In Appendix F we report how the edge ensemble scales in total time for a clearer picture of this drawback. We observe that our link prediction scheme doesn't bring improvements on larger datasets such as ogbn-arxiv(Hu et al., 2020), where the graph is denser and larger, and we do not have the capacity in time to run our link prediction algorithm, as shown in Table 5 in Appendix F. The solution to this problem would be the utilization of other link prediction algorithms, and this is not trivial as most state-of-the-art link prediction algorithms are designed to operate on original full graphs instead of the sub-sampled sparse partitions. However, given the competitive structure certificates we obtained on small to medium graph datasets, we believe that link prediction is essential to achieve better certified accuracies than an MLP in the context of graph structure perturbations.

## 8 RELATED WORK

While robustness certification against test-time attacks is well researched for i.i.d. data as well as for the graph domain (Günnemann, 2022a; Scholten et al., 2022; Hojny et al., 2024), there are few works studying certification against changes to the training data. For the image domain, there are three main approaches: (i) partition-and-aggregate (Levine & Feizi, 2021), (ii) randomized smoothing (over the training data) (Weber et al., 2023), and (iii) differential privacy (Ma et al., 2019); and we refer to Gosch et al. (2025) for a representative survey. Most related to our work is the partition-and-aggregate scheme (Levine & Feizi, 2021), which saw many follow-up works (Wang et al., 2022; Chen et al., 2022; Rezaei et al., 2023). However, it was only applied to the image domain. Regarding graphs, Lai et al. (2024b) develop a probabilistic poisoning certificate against node-injection following the randomized smoothing approach that they extended to collective certification in Lai et al. (2024a), which however is not applicable to the perturbation models studied in this work. Gosch et al. (2025) develops a novel certification paradigm, which is first applied to certify node-feature poisoning and later extended to label poisoning (Sabanayagam et al., 2025), but in both cases is limited to infinite-width GNNs. Further, the label certificate by Sabanayagam et al. (2025) does only scale to datasets having at most 100-200 training labels and thus, does not scale to even our smallest datasets. Li et al. (2025) apply partitioning to derive poisoning certificates for GNNs. However, they do not certify against label flipping, and their structure certificates are below the performance of an MLP (see Appendix E) and thus, vacuous. Further, their feature certification is only applicable to graph classification, as every node is in every partition.

## 9 CONCLUSION

In this paper, we present a self-training framework (ST-GPA) that significantly improves certified robustness against data poisoning in sparse graph-structured data domains. By adding both synthetic labels and structure through effective semi-supervised learning techniques, our method overcomes the limitations of existing partition-based approaches. Empirical results show large improvements in certified robustness to both label and structure poisoning without compromising clean accuracy. Our findings highlight that effectively leveraging semi-supervised learning on sparse data is essential for provably robust graph machine learning against poisoning through partition-based approaches and offer a promising direction for building more robust models beyond the graph domain.

### ETHICS STATEMENT

This work advances the field of certifiable machine learning by enabling robustness of graph neural networks against structure and label poisoning attacks, thereby fostering more reliable and trustworthy machine learning. While there might be many further potential societal consequences of our work, none which we feel must be specifically highlighted here.

### REPRODUCIBILITY STATEMENT

We ensure reproducibility by providing a detailed description of the models, datasets, hyperparameter and random seed selection methods in Appendix A. The code to reproduce our results can be found at: https://figshare.com/s/3beaf03eb7e2c6c8d4ad. Our certification pipeline use only deterministically chosen seeds, so the results are also deterministically reproducible. Alongside the descriptions of our experimental setup we also provide an overview over all parameters in Table 3.

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

## A  EXPERIMENT SETUP

**Datasets.** As decribed in the beginning of Section 6, we use the 3 Planetoid datasets (Yang et al., 2016) Cora, CiteSeer and PubMed, available on pytorch geometric,[3] and the citation dataset Cora-ML (Bojchevski & Günnemann, 2018). As a standard procedure in graph machine learning (Shchur et al., 2018), we preprocess all the datasets by taking the largest connected component and force the graph to be undirected. The statistics of the dataset we use can be found in Table 1. The training, validation and test set nodes are determined with scikit-learn's train_test_split() function,[4] with random_state fixed to 12138. We first separate the test nodes' indices from training and validation (and unused), then use this function again to separate the training set from the validation with the same seed. Only the 30% training labels are available to the model during training. Additionally, we evaluate the label certificates on Wiki-CS (Mernyei & Cangea, 2020) and ogbn-arxiv (Hu et al., 2020), and we use sample 1/10 of their original splits.

Table 1: Statistics of Datasets. The number of training labels consist 30% of all nodes. We adopt a 30%-10%-30% training-validation-test split across all datasets.

| Name | # Nodes | # Training Labels | # Edges | # Features | # Classes | Avg. Degree |
|------|---------|-------------------|---------|------------|-----------|-------------|
| Cora-ML | 2810 | 843 | 7981 | 2879 | 7 | 2.84 |
| CiteSeer | 2110 | 633 | 3668 | 3703 | 6 | 1.74 |
| Cora | 2708 | 812 | 5069 | 1433 | 7 | 1.87 |
| PubMed | 19717 | 5915 | 44324 | 500 | 3 | 2.25 |
| Wiki-CS | 11701 | 586 | 216123 | 300 | 10 | 18.47 |
| ogbn-arxiv | 169343 | 9600 | 1166243 | 128 | 40 | 6.89 |

Table 2: Model Clean Accuracies on Datasets. We report the clean accuracy (percentage) on the test nodes with given dataset and model with the standard deviation of 3 repeated experiments with different initialization seeds to the model.

| | MLP | GCN | GAT | APPNP |
|------|-----|-----|-----|-------|
| Cora-ML | 74.02(0.26) | 78.77(2.10) | 76.95(0.37) | 85.13(0.24) |
| CiteSeer | 70.35(0.20) | 67.46(0.45) | 64.98(2.38) | 72.30(0.61) |
| Cora | 67.78(0.06) | 76.18(0.41) | 75.77(1.22) | 83.39(0.82) |
| PubMed | 73.09(0.05) | 83.66(1.33) | 79.01(0.56) | 87.49(0.41) |

We follow a 30%-10%-30% train-validation-test split of node labels for all experiments following Li et al. (2025). Note that the 10% validation set is typically used for regularization tasks such as hyperparameter tuning and early stopping when training a single GNN. However, since we use a fixed set of hyperparameters and do not employ early stopping, the validation set is not utilized during training.

**Model Parameters.** We used Graph Convolutional Networks (GCN)(Kipf & Welling, 2017), Graph Attention Networks (GAT), and Approximate Personalized Propagation of Neural Predictions (APPNP)(Gasteiger et al., 2018) throughout our evaluations.

Our GCNs consist of 2 layers of GCNConv layer with added self-loops from pytorch geometric.[5] At the bottleneck we use 8 hidden channels, a dropout layer, and ReLU activation. For GAT we use 2 GATConv layers from pytorch geometric.[6] The first layer condense the feature channels to 8 hidden channels with 8 attention heads, and applies dropout and ELU activation, and the second layer then condense the $8 \times 8$ dimension in the middle to class-wise logits. The APPNP model consists of a 2 layer MLP with 8 hidden channels and then an APPNP layer from pytorch geometric[7] to

---

[3]https://pytorch-geometric.readthedocs.io/en/2.6.0/modules/datasets.html

[4]https://scikit-learn.org/stable/modules/generated/sklearn.model_selection.train_test_split.html

[5]https://pytorch-geometric.readthedocs.io/en/2.5.2/generated/torch_geometric.nn.conv.GCNConv.html

[6]https://pytorch-geometric.readthedocs.io/en/latest/generated/torch_geometric.nn.conv.GATConv.html

[7]https://pytorch-geometric.readthedocs.io/en/latest/generated/torch_geometric.nn.conv.APPNP.html

propagate representations, with number of iterations $K = 10$, and teleport probability $\alpha = 0.1$. In structure perturbation experiments specifically, we don't use dropout as it provides us with better results. Additionally, we use an 2 layered MLP as baseline against structure perturbations. It also has 8 hidden units and no dropouts. Note that here in our setup, the GCN with an empty graph is strictly equivalent to the MLP. Both other attack models use dropouts with a probability of $0.5$. All other unstated parameters follow the pytorch geometric default.

**Hyperparameters for Training.** For all GNN training, we use a fixed set of hyperparameters inherited from Li et al. (2018) , which are commonly used in GNN models. Specifically, all models are trained for 200 epochs without early stopping, using the Adam optimizer with a learning rate of 0.01 and a weight decay of $5 \times 10^{-4}$. However, due to the unstable nature of GNN training, we choose the epoch with the lowest loss for prediction. To calculate this without overfitting, we separate the data in each partition in halves, and use one half for training and another for evaluation of the loss. Note that this split happens to the partitioned 30% labels or edges each partition have access to.

For label propagation, we iterate until the score matrix converges, but cap off at 100 iterations.

**Evaluation Process.** As described in Section 5, either all edges, the 30% training set labels, or both are partitioned for the weak classifiers, depending on different types of attacks. We perform transductive node classification, meaning that the weak classifiers then have access to all the graph data $G = (\mathcal{E}, \boldsymbol{X}, \boldsymbol{y})$ except for the poisoned items, which are partitioned in the first place.

All models are evaluated with at least 3 different deterministically chosen seeds for model initialization to evaluate the repeatability of our results. The seeds are chosen by python's random module. We seed the random module with a fixed seed of 123456, and then use random.randint(0, 2**32) to generate a deterministic random seed each time we repeat an experiment.

After the training is done, we evaluate the certified accuracies on the 30% test nodes. This is done by storing the class predictions of each weak classifier for each node and calculate the robust margin as introduced in Equation (8). Then the certified accuracies we reported are given by calculating the ratio of test nodes whose robust margin, together with a specific perturbation size, satisfies Equation (8).

**Hyperparameters Summary.** Here we provide a table of all hyperparameters involved as a summary to our description of the experiment setup, alongside their default values and selection criteria. Throughout this paper, all hyperparameters take their default values (if any) unless otherwise stated.

Table 3: Hyperparameters

| hyperparameter | description | attack models | related models | default value | selection criteria |
|---|---|---|---|---|---|
| lr | learning rate | all | all | 0.01 | follows Li et al. (2018) |
| wd | weight decay | all | all | $5 \times 10^{-4}$ | follows Li et al. (2018) |
| ep | training epochs | all | all | 200 | follows Li et al. (2018) |
| es | early stopping | all | all | None | follows Li et al. (2018) |
| init_seed | seed for python random module | all | all | 123456 | / |
| repetition | number of repeated experiments | all | all | 3 | / |
| num_layers | number of layers in the model | all | all | 2 | prevents overfitting |
| hidden_size | hidden channels | all | all | 8 | prevents overfitting |
| dropout | dropout probability in dropout layers | all | GCN, GAT | 0.5 | prevents overfitting |
| activation | activation function between 2 layers | all | GCN, APPNP, MLP | ReLU | / |
| | | all | GAT | ELU | / |
| train_size | % of labels for training | all | all | 30% | / |
| val_size | % of labels for validation | all | all | 10% | / |
| test_size | % of labels for testing | all | all | 30% | / |
| | | LF | | 80 | Figure 6 |
| $k$ | number of partitions | SP | all | 800 | Figure 11 |
| | | LF&SP | | 200 | Figure 6 |
| order | order of co-training and self-training | LF, LF&SP | all | C/T then S/T | Figure 8 |
| co-train method | ParWalks(PW) or label propagation(LP) | LF, LF&SP | all | LP | Figure 10 |
| $t$ | number of pseudo-labels per class | LF, LF&SP | all | 50 | Figure 13(a) |
| $\alpha$ | teleport probability in label propagation | LF, LF&SP | all | 0.9 | Figure 13(b) |
| $\varepsilon$ | $\times \varepsilon$ pseudo-edges than original graph | SP | all | dataset specific | Figure 14 |

# B ADDITIONAL EXPERIMENT RESULTS

## B.1 CLEAN ACCURACY OF ENSEMBLES

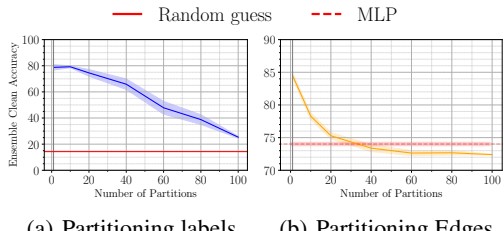

(a) Partitioning labels    (b) Partitioning Edges

Figure 7: Deterioration of clean accuracy of ensembles on Cora-ML. Standard deviations are reported as colored areas over 3 different seeds.

In Figure 7 we show the clean accuracies of an ensemble classifier if the partitioning scheme is applied trivially, i.e. without semi-supervised learning. In the label case, the sparsity degrades the clean accuracy of the ensemble as $k$ goes larger. With $k = 100$ partitions, the clean accuracy drops to just over 20%. In the edge partitioning case, lack of edge information degrades the clean accuracy of an ensemble below that of an MLP's over about $k = 30$ partitions. The deterioration of performance shown here necessitate the introduction of semi-supervised learning schemes within the sparse partitions.

## B.2 ADDITIONAL RESULTS ON OTHER DATASETS

In Figure 5, we presented the proof of 3 statements we made in Section 6 with only results on one selected dataset. Here we present results on other datasets as well and prove that the trends concluded in Section 6 still holds with little further explanations. Note that the subfigures marked with (*) are already present in Section 6.

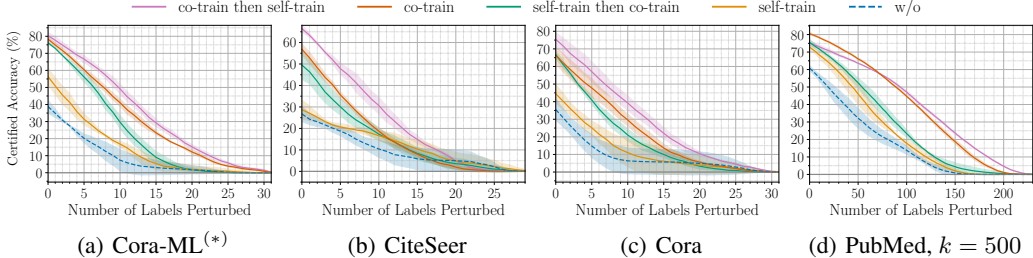

(a) Cora-ML$^{(*)}$     (b) CiteSeer     (c) Cora     (d) PubMed, $k = 500$

Figure 8: We show different orders of co-training and self-training against label flipping on different datasets. $k = 80$ unless otherwise stated.
**The order of co-train then self-train generally works the best against label flipping.** Figure 8 showcases this fact. Our method restores the clean accuracy to generally 70% to 80%, while having higher certified accuracies, despite the poor performance with vanilla partitions.

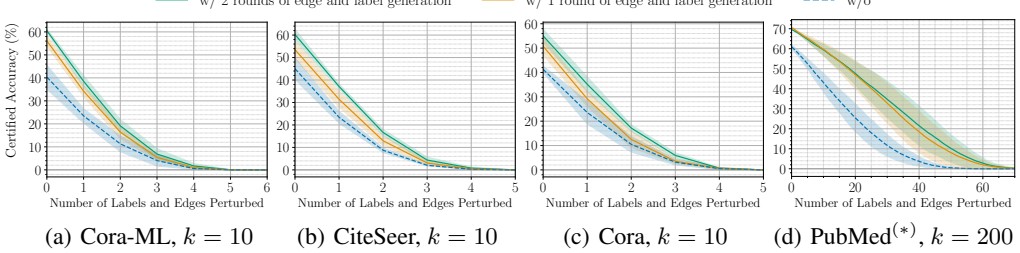

(a) Cora-ML, $k = 10$     (b) CiteSeer, $k = 10$     (c) Cora, $k = 10$     (d) PubMed$^{(*)}$, $k = 200$

Figure 9: Different number of rounds of link prediction + co-training + self-training against both label and structure perturbations.

**Against both label and structure perturbations on smaller datasets, our method provides improvement but the limit of $k$ prevents meaningful joint certificates.** Due to the extreme sparsity of labels and edges if we are facing poisoning of both, we can't use too many partitions, typically limiting $k$ to around 10. Although the curve drops too fast due to a small $k$, yielding hardly any usable certified accuracy against even 1 poisoned label or edge, our method still provides 15% to 20% clean accuracy increase.

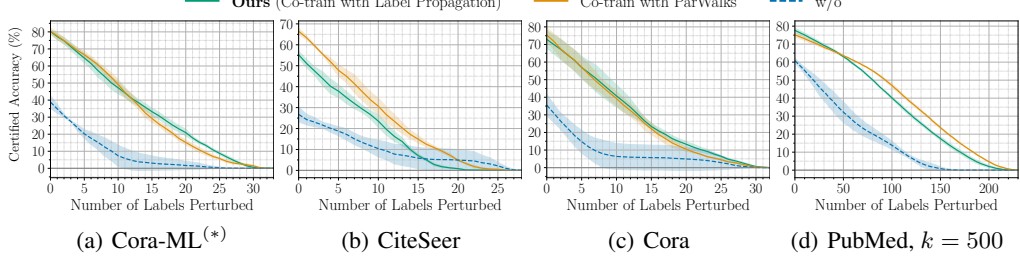

(a) Cora-ML$^{(*)}$     (b) CiteSeer     (c) Cora     (d) PubMed, $k = 500$

Figure 10: Comparison between our method (co-training with label propagation) and ParWalks(Wu et al., 2012). The results are achieved by co-training then self-training on given datasets, with $k = 80$ partitions unless otherwise stated.

**Co-training with label propagation is similarly performant as ParWalks yet scalable.** Our method has similar certified accuracy compared to ParWalks across datasets, except for CiteSeer where it's about 10% less. However, we argue that this is compensated by the vastly shorter training time and the characteristic that scales easily to larger datasets by our method.

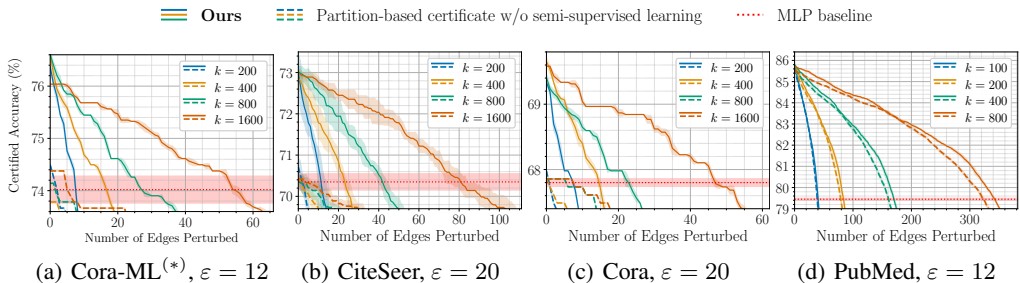

Figure 11: Scalability over $k$ with our link prediction method.

**Link prediction allows arbitrarily large $k$ which generates even better certificates.** In Fig. 11, we demonstrate the near-perfect scalability of link prediction on generating certificates. The number of partitions $k$ determines the maximum achievable robustness with the partitioning scheme. Therefore, given sufficient computational resources, increasing $k$ directly leads to improved robustness. Furthermore, we emphasize that $k$ is theoretically unbounded in its capacity to enhance robustness against structure perturbations.

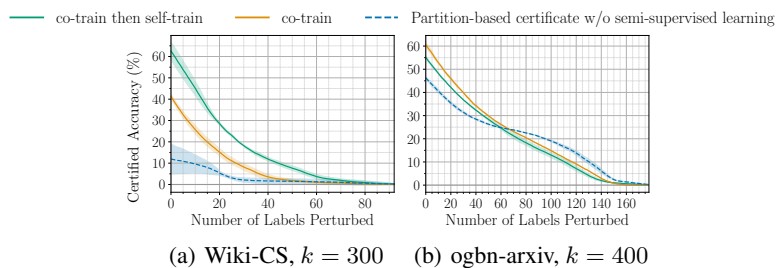

Figure 12: Certifying label flipping on larger graph datasets.

**Our co-training and self-training scheme also scales to even larger datasets.** Figure 12 shows the performance gain upon a trivial partitioning scheme on Wiki-CS and ogbn-arxiv. Our co-training and self-training generally restores a 60% clean accuracy while having also a higher certified accuracy. We used 1/10 label rate as other datasets to demonstrate this improvement with less partitions. If we use the original split, an even larger $k$ would also be possible.

## B.3 ABLATIONS ON HYPERPARAMETERS

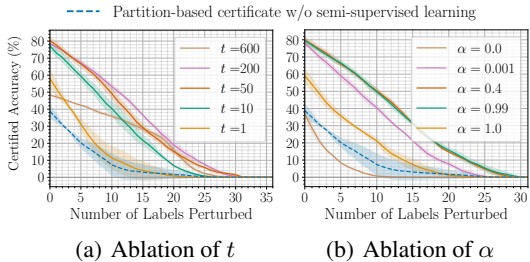

(a) Ablation of $t$      (b) Ablation of $\alpha$

Figure 13: Certifying label flipping on Cora-ML with $k = 80$ partitions.

**The hyperparameter $t$ offers a way to balance between performance and robustness.** Figure 13(a) illustrates the impact of varying the number of pseudo-labels $t$ added during training. When $t$ is small, such as $t = 1$ or $t = 10$, the pseudo-labels are too selective and insufficient in quantity to effectively train the subsequent GNN. Conversely, when $t$ is too large, for example $t \geq 600$, the quality of the pseudo-labels deteriorates, which negatively affects the ensemble classifier's performance. Therefore, selecting an appropriate range for $t$ is critical to achieving optimal certified accuracy. For Cora-ML, this corresponds roughly to the range $50 \leq t \leq 200$. Since $t = 50$ works already quite well and introduce less computation overhead, we fix $t = 50$ for all experiments.

**The random teleport probability $\alpha$ is pretty robust, but need to be properly chosen.** Figure 13(b) shows the certificates with co-training then self-training, but varying the random teleport probability $\alpha$ An $\alpha = 0$ means always randomly teleport to a labeled node in co-training, while an $\alpha = 1$ means no random teleport. As demonstrated by the results, the performance generally plateaus when $0.01 \leq \alpha \leq 0.99$. As long as $\alpha$ isn't chosen to be too large or too small, the certified accuracy remains the best achievable one. We choose to fix $\alpha = 0.9$ for all experiments.

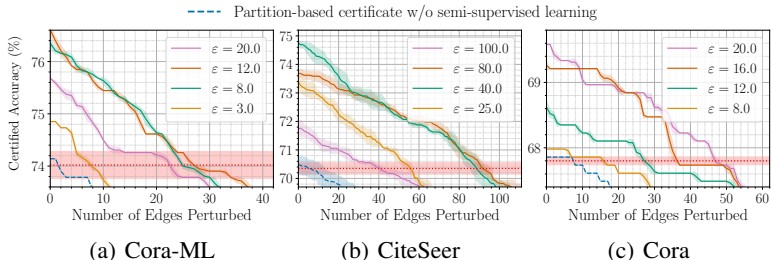

(a) Cora-ML      (b) CiteSeer      (c) Cora

Figure 14: Certifying structure perturbations on the three smaller datasets with $k = 800$ partitions.

**There is a best $\varepsilon$ for each dataset that we can tune to maximize robustness.** $\varepsilon$ serves as a control parameter for the selectiveness in adding pseudo-edges. A too small $\varepsilon$ fails to capture sufficient graph structure, while a too large $\varepsilon$ introduces noise that reduces the signal-to-noise ratio in the generated graph, thereby degrading performance. So there is a best $\varepsilon$ in between, and according to Figure 14, this best value varies a lot between datasets, even though the size of which are similar. To avoid high overheads in all other experiments, we cap $\varepsilon$ to 20.

# C  PROOF OF THEOREMS

## C.1  PROOF OF GENERALIZED DPA

We restate theorem 2 before proving it:

**Theorem 2** (Generalized DPA). *Given a clean, possible non-i.i.d and structured dataset $D$, and a poisoned dataset $\tilde{D}$, the majority-vote classifier prediction remains unchanged, i.e., $g_D(x) = g_{\tilde{D}}(x)$, as long as $d_h d(D, \tilde{D}) \leq \lfloor (n_c(D, x) - \max_{c' \neq c} (n_{c'}(D, x) + \mathbb{1}_{c' < c}))/2 \rfloor = r_m(D, x)$, where $c = g_D(x)$ is the predicted class on the clean dataset.*

*Proof.* We first introduce a lemma that gets rid of the floor operation:

**Lemma 3** (Floor operation equivalency).

$$a \leq \lfloor \frac{b}{2} \rfloor \Leftrightarrow 2a \leq b, \forall a, b \in \mathbb{N} \tag{11}$$

*Proof.* $a \leq \lfloor \frac{b}{2} \rfloor \leq \frac{b}{2} \Rightarrow a \leq \frac{b}{2}$ and $2a \leq b \Rightarrow a \leq \frac{b}{2} \Rightarrow a = \lfloor a \rfloor \leq \lfloor \frac{b}{2} \rfloor$ $\qquad \square$

We use the shorthand notation of $n_c$ being $n_c(D, x)$ and $\tilde{n}_c$ being $n_c(\tilde{D}, x)$, and $r = d(D, \tilde{D})$.

Given Theorem 2 and Lemma 3

$$2d_h \cdot r \leq n_c - \max_{c' \neq c}(n_{c'} + \mathbb{1}_{c' < c}) \tag{12}$$

We get rid of the max operation by taking any class and move all terms to one side

$$0 \leq n_c - n_{c'} - \mathbb{1}_{c' < c} - 2d_h \cdot r, \forall c' \neq c \tag{13}$$

Because the training is conducted in a deterministic manner, the classifier will give the same prediction if the training data is the same. So the number of weak classifiers that predicts differently for a node $f_i(v) \neq \tilde{f}_i(v)$ is also at most $d_h \cdot r$. So for any class, the number of predictions changed is bounded by

$$\forall \bar{c} \in [n_C], |n_{\bar{c}} - \tilde{n}_{\bar{c}}| \leq d_h \cdot r \tag{14}$$

From Equation (14) we plug in $\bar{c} = c$ and $\bar{c} = c'$ to get

$$n_c - \tilde{n}_c \leq d_h \cdot r \tag{15}$$
$$\tilde{n}_{c'} - n_{c'} \leq d_h \cdot r \tag{16}$$

which is equivalent to

$$n_c \leq \tilde{n}_c + d_h \cdot r \tag{17}$$
$$-n_{c'} \leq -\tilde{n}_{c'} + d_h \cdot r \tag{18}$$

Plugging this to Equation (13), we have

$$\forall c' \neq c, 0 \leq n_c - n_{c'} - \mathbb{1}_{c' < c} - 2d_h \cdot r \tag{19}$$
$$\leq \underbrace{\tilde{n}_c + d_h \cdot r}_{\text{Equation (17)}} \underbrace{-\tilde{n}_{c'} + d_h \cdot r}_{\text{Equation (18)}} -\mathbb{1}_{c' < c} - 2d_h \cdot r \tag{20}$$
$$= \tilde{n}_c - \tilde{n}_{c'} - \mathbb{1}_{c' < c} \tag{21}$$

So for the ensemble classifier trained on poisoned data

$$\tilde{n}_c \geq \max_{c' \neq c}(\tilde{n}_{c'} + \mathbb{1}_{c' < c}) \tag{22}$$

and the ensemble classifier's prediction is unchanged because $c$ is the majority. $\qquad \square$

## C.2 Proof of distance function

The following is a proof that $d((\mathcal{E}, \boldsymbol{y}), (\tilde{\mathcal{E}}, \tilde{\boldsymbol{y}}))$ is a distance.

*Proof.* We recall that $d((\mathcal{E}, \boldsymbol{y}), (\tilde{\mathcal{E}}, \tilde{\boldsymbol{y}}))$ is defined to be the sum of symmetric set difference between edges and the hamming distance between labels

$$d((\mathcal{E}, \boldsymbol{y}), (\tilde{\mathcal{E}}, \tilde{\boldsymbol{y}})) = \Delta(\mathcal{E}, \tilde{\mathcal{E}}) + \delta(\boldsymbol{y}, \tilde{\boldsymbol{y}}) \tag{23}$$

and a function $d : M \times M \to \mathbb{R}$ is a distance on metric space $M$ if $\forall x \in M$, $d(x, x) = 0$, $d(x, y) > 0, x \neq y$, $d(x, y) = d(y, x)$, and $d(x, y) \leq d(x, z) + d(y, z)$.

We first note that $d(\cdot, \cdot)$ between one $(\mathcal{E}, \boldsymbol{y})$ and itself is $0$ because both its terms are $0$; It's value is always positive as it's the sum of two positive numbers, if both input are distinct; $d(\cdot, \cdot)$ is symmetric because both its terms are symmetric and changing the order of objects doesn't affect the value.

To prove triangle inequality

$$d((\mathcal{E}_1, \boldsymbol{y}_1), (\mathcal{E}_3, \boldsymbol{y}_3)) = \Delta(\mathcal{E}_1, \mathcal{E}_3) + \delta(\boldsymbol{y}_1, \boldsymbol{y}_3) \tag{24}$$
$$\leq \Delta(\mathcal{E}_1, \mathcal{E}_2) + \Delta(\mathcal{E}_2, \mathcal{E}_3) + \delta(\boldsymbol{y}_1, \boldsymbol{y}_2) + \delta(\boldsymbol{y}_2, \boldsymbol{y}_3) \tag{25}$$
$$= d((\mathcal{E}_1, \boldsymbol{y}_1), (\mathcal{E}_2, \boldsymbol{y}_2)) + d((\mathcal{E}_2, \boldsymbol{y}_2), (\mathcal{E}_3, \boldsymbol{y}_3)) \tag{26}$$

So $d((\mathcal{E}, \boldsymbol{y}), (\tilde{\mathcal{E}}, \tilde{\boldsymbol{y}})$ is indeed a distance metric.

$\square$

---

**Algorithm 3** Efficient Edge Candidate Selection for Link Augmentation

---

**Require:** Number of nodes per class $n_c$, node confidence scores per class:
    $\{S_c = [s_{c,1}, s_{c,2}, \ldots, s_{c,n_c}]\}$ sorted descending, desired number of edges to add $\varepsilon \cdot e$
**Ensure:** Top-$\varepsilon \cdot e$ pseudo-edges with highest sum of confidence scores
  1: Initialize global max-heap $G$ (size = number of classes)
  2: **for** each class $c$ **do**
  3:    Initialize empty local max-heap $L_c$
  4:    Insert initial pair $(0, 1)$ with score $s = S_c[u] + S_c[v]$ and class index as tuple $(u, v, s, c)$ into
      $G$
  5: **end for**
  6: **while** number of added edges smaller than $\varepsilon \cdot e$ **do**
  7:    Extract top $(u, v, s, c)$ from global heap $G$
  8:    Add $e = (u, v)$ to candidate edge set
  9:    For local heap $L_c$, consider children pairs:
10:      $child_1 = (u, v + 1)$ if $v + 1 \leq n_c$
11:      $child_2 = (u + 1, v)$ if $u + 2 = v$
12:    **for** each valid child pair $(u, v)$ **do**
13:      Compute sum of score $s = S_c[u] + S_c[v]$
14:      Insert tuple $(u, v, s)$ into $L_c$
15:    **end for**
16:    Extract next top tuple $(u', v', s)$ from $L_c$
17:    Insert $(u', v', s, c)$ into global heap $G$
18: **end while**
19: **return** Final candidate edges across all classes

---

# D ALGORITHMS

## D.1 EFFICIENT SELECTION OF PSEUDO-EDGE CANDIDATES

As the number of nodes, $k$ or $\varepsilon$ goes up, the number of possible edge pairs also scales up, which results in significant computational overhead. To address this challenge, we propose a time and memory-efficient algorithm for selecting edge candidates with the highest combined confidence scores across all classes. The algorithm for selecting edges is described in Algorithm 3. A global max-heap keeps track of which class has the next best edge candidate according to the sum of scores. We initialize the global heap by inserting the first pair $(0, 1)$ from each class, as these pairs hold the maximum possible sum of scores within their respective classes. To add edges, we repeatedly pick the top element from the global heap until the number of edges selected exceeds the threshold defined by $\varepsilon$.

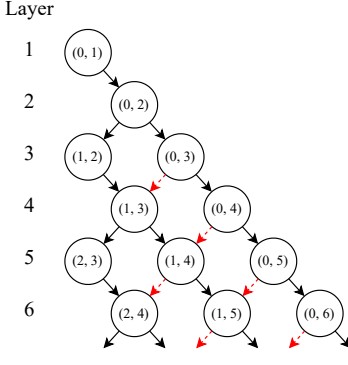

Figure 15: Tree structure for spawning edge candidates. An edge in layer $i$ is guaranteed to have a higher score than in layer $i + 1$ and a lower score than in layer $i - 1$.

To supply the global heap with the best candidates from each class, every class maintains a local heap, which is initially empty. Each time an edge is selected from the global heap, the algorithm accesses the corresponding class's local heap and inserts the next candidate pairs. To prevent local heaps from becoming prohibitively large, we do not insert all possible node pairs at once. Instead, node pairs are added only when they can potentially represent the best candidate. This is enabled by pre-sorting nodes and their confidence scores within each class in descending order. Consequently, nodes with smaller indices correspond to higher scores. This ordering induces a binary tree structure over node pairs, illustrated in Figure 15, where an edge $(u, v)$ has a larger combined score than $(u, v + 1)$, and $(u + 1, v)$. As the criterion, the sum of scores, is commutative, $(u, v)$ and $(v, u)$ represent the same edge candidate, so we consider only node pairs $(u, v)$ where $u < v$. This results in the half binary tree structure in Figure 15.

Since most nodes have two parents, simply adding both $(u, v+1)$ and $(u+1, v)$ to the local heap would result in duplicate insertions of many nodes. To avoid this, we add the pair $(u+1, v)$ only when $u+2 = v$. This condition effectively removes the red dashed connections shown in Figure 15 and ensures that each edge pair is added to the local heap exactly once. This also guarantees that the local heap always contains the node pair with the highest possible sum of scores.

### D.2 PROPOSED GENERAL CERTIFICATION PIPELINE AGAINST BOTH LABEL FLIPPING AND STRUCTURE PERTURBATIONS

---
**Algorithm 4** ST-GPA joint pipeline

---
**Require:** Graph dataset $D = (\mathcal{E}, \mathcal{X}, \mathcal{Y})$, a label propagation method $\hat{\boldsymbol{y}} = LP(\varepsilon, \mathcal{X}, \boldsymbol{y})$, a label self-training method $\hat{\boldsymbol{y}} = LST(\varepsilon, \mathcal{X}, \boldsymbol{y})$, an link prediction method $\hat{\varepsilon} = EST(\varepsilon, \mathcal{X}, \boldsymbol{y})$, a training order $o_t = \{LP, LST, EST\}^m$
**Ensure:** A robust ensemble classifier $g$
1: split poisoned targets into partitions $\varepsilon_i = \{e \in \mathcal{E} | h(e) \equiv i\}$ and/or $\boldsymbol{y}_i = \{-1\}^x \cup labels$
2: **for** each operation $op$ in the given order $o_t$ **do**
3:   **for** each partition $i$ **do**
4:     $\hat{\varepsilon}$ or $\hat{\boldsymbol{y}} = op(\varepsilon_i, \mathcal{X}, \boldsymbol{y}_i)$
5:     $\varepsilon_i = \varepsilon_i \cup \hat{\varepsilon}$
6:     $\boldsymbol{y}_i = \boldsymbol{y}_i \cup \hat{\boldsymbol{y}}$
7:   **end for**
8: **end for**
9: train weak classifiers on final edges, labels and features
10: count weak classifier predictions
11: calculate certificate

---

# E DISCUSSION ON THE RELATION TO PGNNCERT AGAINST STRUCTURE PERTURBATIONS

Li et al. (2025) apply a similar partitioning scheme to derive poisoning certificates for GNNs. In their work, a thread model of arbitrarily perturbing edges, nodes, and node features is considered. In this section, we provide comparison between our results and theirs. Due to their certified accuracy does not outperform an infinitely robust MLP, we do not report them in Section 6.

To compare our works directly, we adopted their published code and use their exact dataset splits. We then run our proposed approach on their data split. Note that the only difference in the setup is the model and hyperparameters for training, where we use a 2-layered GCN as described in Appendix A, and they use a 3-layered GCN with skip connections and linear layer. In Li et al. (2025), 2 different partitioning schemes were proposed, namely edge centric and node centric, which has similar performance against structure perturbations, so we report the result with the node centric variant only. Due to the limitation of implementation of Li et al. (2025), their code does not scale to larger number of partitions due to memory limits, so we choose $k = 60$ as a compromise and keep other parameters exactly the same for a fair comparison. Note that normally our link prediction scheme can be used with significantly larger $k$-s to generate competitive certificates, so we also report $k = 600$ with the same setup. Although Li et al. (2025) doesn't report results on Cora, their code can be easily adapted as the Planetoid datasets share the same data loader in PyTorch, so we report results on Cora as well.

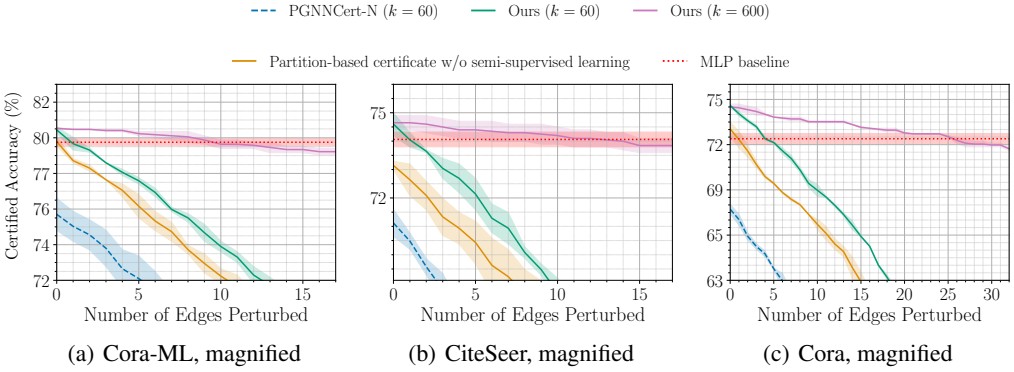

(a) Cora-ML, magnified      (b) CiteSeer, magnified      (c) Cora, magnified

Figure 16: Certified accuracies of our method compared to Li et al. (2025), $\varepsilon = 30$. PGNNCert-N's performance falls steadily below the MLP baseline which is infinitely robust to structure perturbation; a trivially applied partitioning scheme has only on-par clean accuracy with an MLP, yielding hardly any meaningful perturbation budget; with our proposed link prediction method, the certified accuracies is most competitive.

As shown in Figure 16, PGNNCert-N's performance is consistently outperformed by the infinitely robust MLP baseline on all datasets. In the mean time, our link prediction method shows similar improvements to certified accuracy over the MLP baseline as previously reported in Section 6 and Appendix B. At $k = 60$, the allowed perturbation budget is very small. But the reported $k = 600$ curves shows the normal performance of our method with more partitions, yielding larger perturbation budgets.

Provided that PGNNCert's performance is below the MLP baseline in the case of structure perturbation, we don't report a comparison to PGNNCert in the results in this paper, as we already use the MLP baseline as a lowest acceptable case in all our results against structure perturbation.

In Li et al. (2025), no certificate against label flipping is reported. As we have already shown in Appendix B.1, the partitioning scheme does not readily transfer to labels, so we don't compare our results with Li et al. (2025) on label flipping.

# F  SCALABILITY OF PROPOSED METHODS

In this section, we analyze the scalability of our 3 proposed self-training methods by reporting the relative time used in our experiments. In all our experiments, we use one single NVIDIA GTX1080Ti GPU. The link prediction uses CPU only. While it can be easily parallelized, we report the performance on 1 CPU core only, as link prediction on individual partitions are usually done so. The time reported are for relative reference considering the number of nodes and edges in each dataset.

We first point out that the time complexity scales linearly with the number of partitions $k$, since the partitions are disjoint once they are created, Partitioning the dataset takes a negligible amount of time, and can easily be accelerated by pre-processing the dataset and store all the partition indices. Consequently, it is reasonable that we only report time taken per partition in this section. The memory complexity is irrelevant w.r.t. $k$ by the same reason that the partitions are disjoint. The training on individual partitions could be done in a parallel or distributive manner.

Table 4: Time used for co-training and self-training with $t = 50$

|  | Cora-ML | CiteSeer | Cora | PubMed | Wiki-CS | ogbn-arxiv |
|---|---|---|---|---|---|---|
| graph size (Nr. nodes) | 2810 | 2110 | 2708 | 19717 | 11701 | 169343 |
| **training time per partition (s)** | **6.4** | **5.5** | **5.5** | **2.7** | **3.1** | **40.9** |
| thereof training w/o SSL | 36.1% | 37.3% | 36.4% | 31.9% | 29.0% | 31.4% |
| thereof C/T | 26.8% | 24.7% | 25.2% | 30.8% | 36.4% | 37.0% |
| thereof S/T | 37.1% | 38.0% | 38.4% | 37.3% | 34.6% | 31.6% |

Table 4 shows the time taken to train an individual partition on all datasets we tested. Label partition training is generally very fast. Note that the time portion reported for training without semi-supervised learning represents the time baseline for training on the raw partition, which has also similar time complexity of training on the clean graph. Judging from the percentages reported, co-training and self-training is just another round of training of a GNN, which takes similar time as training the first one on the raw partition. The results show good scalability of our label self-training methods as the time portion stays roughly equal regardless of the size of the dataset, and the total time necessary to train the ensemble scales roughly linearly with the graph size (number of nodes).

Table 5: Time used for link prediction with $\varepsilon = 1.0$

|  | Cora-ML | CiteSeer | Cora | PubMed | Wiki-CS | ogbn-arxiv |
|---|---|---|---|---|---|---|
| graph size (Nr. edges) | 7981 | 3668 | 5069 | 44324 | 216343 | 1166243 |
| **training time per partition (s)** | **1.9** | **1.0** | **2.0** | **9.3** | **19.3** | **67.8** |
| thereof training w/o SSL | 32.3% | 58.7% | 57.5% | 14.4% | 8.2% | 6.6% |
| thereof L/P | 7.3% | 3.8% | 5.5% | 13.8% | 34.5% | 42.8% |
| thereof training with pseudo-edges | 60.4% | 37.5% | 37.0% | 71.8% | 57.2% | 50.6% |

Table 5 shows the time taken per partition for edge semi-supervised training. Although smaller datasets has faster training time per partition, more partitions are required in structure perturbation to generate a meaningful certificate. Therefore, the time taken to train an edge ensemble is usually comparably longer than an label ensemble. In the smaller datasets, training without and with the pseudo-edges take roughly the same time. However, as the number of edges grows, training with pseudo-edges becomes the predominant factor in total training time, and the time taken to find the pseudo-edges becomes non-negligible. On larger graphs, this is especially a choking factor as we usually add more pseudo-edges than there originally are ($\varepsilon \gg 1.0$), making the link prediction very time-consuming.

## G  CLEAN ACCURACY INCREASE WITH SEMI-SUPERVISED LEARNING

Our semi-supervised learning methods are tailored to boost *certifiable* robustness by improving the weak classifier's performance. Compared to a single classifier, the semi-supervised learning methods we propose are much more powerful on weak classifiers. By doing co-training, self-training and link prediction, the weak classifiers' performance gets boosted, which in turn yields improved certifiable robustness for the ensemble.

When compared to a boost of accuracy of the weak classifier, an ensemble classifier's clean accuracy increases by a larger margin. This is due to the nature of an ensemble, where only a majority of votes has to be correct. This is shown in Table 6. An ablation that how semi-supervised learning performs without partitioning is also included for comparison.

Table 6: Clean accuracies (%) of weak classifiers and ensembles on Cora-ML

|  |  | w/o | C/T | S/T | L/P |
|---|---|---|---|---|---|
| $k = 1$ (brings no robustness) |  | 78.77 | 80.39 (+1.52) | 78.92 (+0.15) | 78.99 (+0.22) |
| partition labels, $k = 80$ | individual | 25.80 | 39.91 (+14.11) | 43.45 (+17.65) |  |
|  | ensemble | 38.79 | 76.59 (+37.80) | 80.70 (+41.91) |  |
| partition edges, $k = 800$ | individual | 71.28 |  |  | 70.32 (-0.96) |
|  | ensemble | 74.26 |  |  | 76.34 (+2.08) |

Individual classifiers' accuracy is the average value over all partitions.

As shown in Table 6, the clean accuracy increase is little, if any, when applied to the unpartitioned dataset. The boost of clean accuracy is the most effective on partitioned data, where the baseline accuracy is low due to sparsity. In general, the ensemble classifier's accuracy increases by a larger margin compared to individual classifiers'. In the structure perturbation case, the weak classifiers even observe a decrease in accuracy, but the ensemble has nonetheless a higher clean accuracy. This is due to that more samples have a majority of correct predicting weak classifiers after link prediction.

The results further strengthen the necessity of our proposed semi-supervised learning scheme. While the partitioning provides the fundamental provable robustness, the individual classifiers perform poorly under sparse conditions in the vanilla setup. Here, our proposed semi-supervised training scheme is fully responsible for the improvement of the weak classifiers' accuracies, in turn resulting in much better clean and certified robustness for the overall ensemble.

# H LLM USAGE

LLMs are used to assist with writing tasks such as grammar checking and polishing. They are not employed in the research ideation process or in generating the text in this paper.

