# OpenReview forum: "Certifying Graph Neural Networks Against Label and Structure Poisoning"
_ICLR.cc/2026/Conference — Submitted to ICLR 2026_

### Official Review · Reviewer_Zr4m · 2025-10-19

**Soundness:** 2
**Presentation:** 3
**Contribution:** 2
**Rating:** 2
**Confidence:** 3

**Summary:**

Partition-based certification methods struggle to provide meaningful poisoning guarantees for GNNs because partitioning graph data often creates overly sparse subgraphs for training base models. To overcome the limitations, this paper introduces ST-GPA, using semi-supervised learning and link prediction to generate pseudo-labels and synthetic edges to densify each partitioned subgraphs. Experimental results show that ST-GPA consistently improves robustness guarantees against both label and structure poisoning across multiple dataset.

**Strengths:**

* The writing is well-structured and easy to follow.
* Addresses an important problem in guaranteeing robustness against poisoning attacks
* Their method certifies defense against both node-level and structure-level attacks.

**Weaknesses:**

* The authors state that their method maintains strong clean accuracy. However, their figures do not provide any comparison against clean accuracy of vanilla GNNs without any partitioned training. The authors should integrate the details from Table 2 (appendix) in the main paper for better clarity.

* The author only evaluate under citation-based dataset with the largest dataset being medium sized (Pubmed). I highly encourage the authors to provide evaluations results on large scale dataset such as arXiv and non-citation dataset like WikiCS.

* The core certification guarantee (Theorem 2) is a minor generalization of the existing DPA theorem and simplifies (d_h=1) in the paper's specific application, making the resulting condition formally analogous to the original DPA framework.

* While the reasons for omission of certain baselines are described in section 7, [1] should still be included as a baseline. Although [1] doesn't report certificates against label flipping, its underlying partition-based methodology is general and applicable.

**Questions:**

* In the proof for Theorem 2, shouldn't the minus be a plus in equation 22?

* Clarification is needed regarding the claim that the structural certificates from [1] fall below MLP performance. There appears to be no direct evidence supporting this assertion presented either in the current paper or in the cited work [1]

\
[1] Deterministic certification of graph neural networks against graph poisoning attacks with arbitrary perturbations.

---

> ### Author Response · Authors · 2025-11-24
> **Response to Reviewer Zr4m**
>
> We sincerely thank the reviewers for their thoughtful and constructive feedback. We have carefully considered the points raised and provide detailed responses below addressing the concerns and questions.
>
> **Weakness 1 (Vanilla GNN accuracy comparison)**
>
> In response to the reviewer's comment, we include the clean accuracies in Figure 4 (Page 7) and also link to Table 2 in the main text. We additionally clarified in the text that our method achieves significantly stronger clean accuracies when compared to previous partitioning-based robustness certificates for graph neural networks. Beyond that, please note that the performance of vanilla GNNs does not hold under adversarial attacks. In contrast, our approach provides a lower bound on the accuracy that provably holds even under adversarial attacks.
>
> **Weakness 2 (Larger datasets)**
>
> We have now added an evaluation of our label certificates on Wiki-CS and ogbn-arxiv in Fig. 12 in Appendix B.2 (Page 17). The results show that our co-training and self-training scheme improves upon the vanilla partitioning scheme and generates strong certified accuracies against label poisoning. Our link prediction scheme struggles on Wiki-CS as well as ogbn-arxiv. This has two reasons: (i) For Wiki-CS, neighbourhoods are significantly more heterophilic, complicating the link-prediction task; (ii) link prediction is more expensive than label propagation making scaling to datasets with more than tens of thousands of nodes difficult. We acknowledge and discuss these two limitations of our method regarding structure certificates in detail in our discussion in Section 7 (Page 9).
>
> **Weakness 3 (Similarity to DPA)**
>
> We acknowledge that Theorem 2 is a minor but important generalization of DPA by showcasing how to formally derive the DPA condition with more complex partitioning schemes on non-i.i.d. structured data. Thus, our contribution in that respect is a more general formulation of the original DPA proof. However, our *main contribution* lies in the introduction of semi-supervised label and structure augmentation, which is essential to make partitioning based certificates work in the graph domain. Vanilla DPA fails in graphs, where labels or edges are sparse. If DPA is trivially applied to graphs, the label certificate would be very low (unusable in real world applications), and the structure certificate would not outperform an MLP, which doesn't use edges at all. This is proven by the low clean accuracies we report in Figure 7 in Appendix B.1 (Page 15), as any certified accuracy is not higher than the clean accuracy in our partitioning scheme. Our contribution would be to not only extend DPA effectively to the graph domain, but also make GNNs work on the partitioned graph data to yield meaningful certificates.
>
> **Weakness 4 (A comparison to Li et al.)**
>
> We now compare to Li et al. [1] in Fig. 16 in Appendix E (Page 23). It shows that our method significantly outperforms their method, and also that the structural certificate from [1] falls below an MLP baseline, whereas our method significantly improves upon it.
>
> We want to note that we compare the structure certificates. We did not compare certifying label poisoning, as [1] did not apply their certificate to label poisoning, and the partitioning scheme from [1] is not directly applicable to certify label poisoning in node classification tasks, as the labels are included in each partition and thus, one poisoned label can affect all partitions leading to a trivial certified accuracy of 0%.
>
> [1] Li et al. "Deterministic certification of graph neural networks against graph poisoning attacks with arbitrary perturbations", CVPR 2025
>
> **Question 1 (Typo in proof)**
>
> Yes, and we have corrected it. We thank the reviewer for spotting it. Please note that this does not change our theoretical and experimental results and was just a minor typo.
>
> **Question 2 (Structural certificates from Li et al. below MLP)**
>
> In response to the reviewer's comment and as mentioned above, we now report the structural certificates from Li et al. in Fig. 16 in Appendix E (Page 23). The evaluation is done using the code published in their repository, and the results show that the certificates fall below MLP performance. Our initial statement was motivated by having the same experimental setup as Li et al. (i.e., the same train-test split percentages) and comparing their reported certified accuracies from their paper with our observed MLP baseline performance.

---

### Official Review · Reviewer_m8sV · 2025-10-31

**Soundness:** 2
**Presentation:** 3
**Contribution:** 2
**Rating:** 4
**Confidence:** 3

**Summary:**

The paper tackles the problem of certifying robustness of Graph Neural Networks (GNNs) against training-time poisoning attacks on node labels and graph structure. It extends the Deep Partition Aggregation (DPA) framework to non-i.i.d. graph data and introduces Self-Training Graph Partition Aggregation (ST-GPA), which enriches each sparse partition with pseudo-labels and pseudo-edges via semi-supervised learning. Experiments on multiple benchmarks show that ST-GPA substantially improves both clean accuracy and certified robustness, establishing an effective certification framework for label- and structure-poisoned GNNs.

**Strengths:**

1. The paper targets training-time attacks and provides a clear certified condition based on the robust margin.
2. The work extends Deep Partition Aggregation (DPA) from i.i.d image data to non-i.i.d. graph data.
3. Evaluations across multiple benchmarks (Cora, CiteSeer, PubMed, Cora-ML) and architectures (GCN, GAT, APPNP) show consistent certified accuracy improvements.

**Weaknesses:**

1. Self-training has already been shown to significantly improve adversarial robustness in GNNs[1,2]. Therefore, I strongly suspect that the robustness improvement observed in this paper may mainly stem from the self-training component, rather than from the proposed Graph Partition Aggregation mechanism itself.
2. The paper lacks comparisons with strong existing certified-robustness baselines, such as NTK-based or MILP-based exact certificates[3].

[1] Li, Kuan, et al. "Revisiting graph adversarial attack and defense from a data distribution perspective." The Eleventh International Conference on Learning Representations. 2023.

[2] Chowdhury, Subhajit Dutta, et al. "Unveiling Adversarially Robust Graph Lottery Tickets." Transactions on Machine Learning Research.

[3] Gosch et al., 2024 — Provable Robustness of (Graph) Neural Networks Against Data Poisoning and Backdoor Attacks

**Questions:**

1. Can ST-GPA apply to larger graphs (e.g., ogbn-arxiv, ogbn-products, ogbn-papers100M) and how's the certified accuracy?
2. It's always worth to see a defense method's robustness under adaptive attacks: Consider scenarios where an attacker knows the pseudo-labeling mechanism.

---

> ### Author Response · Authors · 2025-11-24
> **Response to reviewer m8sV**
>
> We thank the reviewers for their detailed critiques. We have thoroughly reviewed the concerns and provide our point-by-point responses below.
>
> **Weakness 1 (Existing self-training defenses)**
>
> We would like to highlight that the mentioned works are empirical defenses and our method is a robustness certificate, which means that the certified accuracies we report are mathematically proven lower bounds against any attack. Therefore, this paper does not try to improve the robustness achieved by self-training in mentioned works, but generalize deep partition aggregation (DPA)[1], the state of the art certifiable defense to data poisoning (on image datasets), to graphs. This proposed Graph Partition Aggregation backbone is essential to derive our certificates. Self-training, in our work, is used as an essential tool that leverages the graph structure to improve the individual classifiers' accuracies, achieving competitive certified accuracies for the ensemble classifiers. In short, the Graph Partition Aggregation provides a foundation for generating **certified** accuracies, and our proposed semi-supervised learning methods improve upon the Graph Partition Aggregation to yield meaningful certified accuracies.
>
> [1] A Levine and S Feizi. Deep partition aggregation: Provable defense against general poisoning attacks. ICLR 2021
>
> **Weakness 2 (Existing certificates)**
>
> We assume the reviewer is referring to LabelCert introduced by [1] as the referenced work by Gosch et al. [2] only deals with feature poisoning and thus, is not applicable to our work, where we derive a certificate for label and structure poisoning. Indeed [1] is a very interesting work and we wanted to compare against their results, but this was hindered by the strong scaling limitations of LabelCert. In particular, the method of [1] does not scale to even the smallest of our instance sizes. The reason is that LabelCert can only deal with datasets having a total of at most 100-200 training labels. Our smallest dataset w.r.t. training labels in the submitted draft was CiteSeer with 633 training labels, and in the revised draft Wiki-CS with 586 training labels. In our training-test splits, we followed Li et al. [3] for comparability purposes with their work (see the newly added Appendix E, Page 23 for explicit comparisons to Li et al.) and as it represents a sensible split for partition-based approaches. We now discuss this point in more detail in our related work on page 10.
>
> [1] Sabanayagam et al., Exact Certification of (Graph) Neural Networks Against Label Poisoning, ICLR 2025
>
> [2] Gosch et al., Provable Robustness of (Graph) Neural Networks Against Data Poisoning and Backdoor Attacks, AdvML @ NeurIPS 2024
>
> [3] Li et al. Deterministic certification of graph neural
> networks against graph poisoning attacks with arbitrary perturbations, CVPR 2025
>
> **Question 1 (Larger graphs)**
>
> We now include the label certificates on ogbn-arxiv in Fig. 12 in Appendix B.2 (Page 17). On Wiki-CS, our co-training and self-training boost clean accuracy from lower than 15% to more than 60%; on ogbn-arxiv, co-training introduces a 15% increase to clean accuracy to 60% while improving the certified accuracy up to 60 label flips. The improvement of clean accuracy and certified robustness proves the effectiveness of our co-training and self-training scheme, and they scale very well to larger graphs.
>
> Our link prediction method, on the other hand, scales less favorably due to the many pseudo-edges generated. We acknowledge this now as a limitation and include it in the discussion in Section 7 (Page 9). We also want to note that we believe that improving link prediction step through an even more targeted pseudo-edge generation has the potential to significantly speed up the structure certificate making it too, scale well to large graphs, which we think can be an interesting avenue for future study.
>
> **Question 2 (Adaptive attacks)**
>
> We would like to clarify that our method provides provable robustness guarantees. This means that the certified accuracies provided by our method are mathematically proven lower bounds against **any** type of attack within the considered attack budget (see Theorem 2, Page 4). Thus, our reported certified accuracies are also valid under any thinkable adaptive attack that knows the pseudo-labeling mechanism. Our reported results can be interpreted as "The best an adversary could do, even if it knows the partitioning scheme and the semi-supervised training algorithm, is to reach our provided lower bound." This is in contrast to empirical defenses, which only provide empirical statements of robustness gains and have to be empirically tested against adaptive attacks. However, our method, as a certified defense, provably includes any attack possible, as our guarantees are proven to hold against the worst-case attack possible in the given attack budget and thus, inherently already include the asked for adaptive attacks.

---

> > ### Comment · Reviewer_m8sV · 2025-11-26
> >
> > Thanks the author for the rebuttal and the additional experimental results. The new comparisons with Li et al. (CVPR 2025), the inclusion of Wiki-CS and ogbn-arxiv for label certification, and the scalability analysis help clarify several of my earlier concerns.
> >
> > One remaining point I would like to discuss is still about the source of robustness gains. As the author emphasized, ST-GPA is a certified defense and therefore does not require empirical attack evaluations to claim provable guarantees. At the same time, nearly all improvements in certified accuracy arise from the fact that self-training and pseudo-edge augmentation substantially strengthen each base classifier, which in turn increases the voting margin. Since the paper does not include empirical poisoning experiments, it is difficult to disentangle how much of the overall robustness—beyond the formal certificate itself—comes from the partition-aggregation mechanism versus from the self-training components.
> >
> > Indeed, I think maybe one additional ablation would help clarify this distinction: applying the same self-training pipeline (label propagation, self-training, and pseudo-edge generation) without partitioning, i.e., training a single GNN with the SSL augmentation but no partition-aggregation step. Even a lightweight version of this experiment on one dataset like Cora-ML would make it easier for me to understand how much of the robustness improvement is attributable to the SSL components alone and how much comes from making partition-based certification viable on sparse graphs.

---

> > > ### Author Response · Authors · 2025-12-02
> > >
> > > Thank you for your continued feedback. In response, we added results reporting the clean accuracy increase without any partitioning (Table 6, Page 25). We would like to emphasize that our work focuses on *certifiable* robustness, whose goal is to provide *provable* lower bounds on robustness. Importantly, partitioning is essential for achieving any non-trivial robustness guarantees: without data partitioning, the predictions are not certifiably robust (even under a single label or edge perturbation). In this context, our proposed self-training approach is fully responsible for the improvements in *certifiable robustness* when compared to the baselines in Fig. 4 (Page 7) and Fig. 5–6 (Page 8).
> > >
> > > We thank the reviewer again for the constructive comment and hope this clarifies the distinction between empirical and certifiable robustness.

---

### Official Review · Reviewer_qsA6 · 2025-10-31

**Soundness:** 2
**Presentation:** 3
**Contribution:** 2
**Rating:** 4
**Confidence:** 3

**Summary:**

This paper proposes a model that achieves robustness for Graph Neural Networks (GNNs) against label and structure poisoning. It effectively describes the challenges of applying certification methods to graphs, which differ significantly from the image domain. The proposed model utilizes a self-training-based ensemble of classifiers to make robust predictions and demonstrates consistent performance improvements across various datasets.

**Strengths:**

S1. The paper addresses the vulnerability of GNNs to adversarial poisoning attacks, which is a critical and still largely unresolved problem in the field.

S2. It clearly demonstrates the limitations of the naive partitioning-and-aggregation method (which works for images) when applied to graphs, supporting this analysis with experimental results (e.g., Figures 2 and 3).

S3. The proposed method, ST-GPA, shows consistent and significant performance improvements over the baselines on multiple datasets and GNN architectures.

**Weaknesses:**

W1. The motivation for the work could be strengthened and more clearly articulated in the introduction, compared to the existing work. In my opinion, the discussion on self-training for GNN robustness could be more comprehensive. There are several relevant studies, such as Good-at [1] and GPR-GAE [2], that are not discussed. A discussion of how the proposed method relates to these and, if possible, an experimental comparison would be beneficial.

W2. The evaluation of the adversarial setting should be extended to more realistic and challenging attack scenarios, such as those generated by optimization-based algorithms (e.g., PR-BCD, LR-BCD). The current evaluation assumes a simpler threat model, which may limit the perceived benefits and generalizability of the proposed method.

W3. The paper claims that the guarantee from Theorem 1 (DPA) "does not depend on the i.i.d. nature of the data." in Lines 140-141. This claim needs further clarification. How is the i.i.d. assumption, which is fundamental to the original DPA proof (as poisoning one data point only affects one partition), properly bypassed in the generalized non-i.i.d. graph setting?

W4. A significant limitation is that the pseudo-edge generation technique (Section 5.2) is explicitly "guided by the homophily assumption." This approach is likely to be ineffective or even detrimental on heterophilous graphs, where connected nodes often have different labels. This limitation should be acknowledged and discussed.

W5. While the paper is generally well-written, there are areas for improvement in its presentation. For example,

- Some notations are unclear (e.g., the variable '$r$' in Theorem 1 is used without a clear definition in that immediate context).

- The font size in Figure 1 is too small, which hinders readability.


References

[1] Boosting the adversarial robustness of graph neural networks: An ood perspective, ICLR 2024

[2] Self-supervised Adversarial Purification for Graph Neural Networks, ICML 2025

**Questions:**

Q1. Could you strengthen the motivation by differentiating more clearly from existing work?

Q2. The current evaluation uses a basic threat model. To what extent would the certified guarantees hold against more realistic, optimization-based attacks like PR-BCD or LR-BCD?

Q3. Could you please clarify the claim in Lines 140-141? How is the i.i.d. assumption from the original DPA proof (where one poisoned point affects one partition) provably bypassed in your generalized non-i.i.d. graph setting?

---

> ### Author Response · Authors · 2025-11-24
> **Response to reviewer qsA6 (Part 1)**
>
> We thank the reviewers for their detailed and in-depth review. We address all the weaknesses and questions raised by the reviewer below.
>
> **Weakness 1 & Question 1 (Other self-training defenses)**
>
> Firstly, we would like to emphasize the domain difference between empirical defenses and certified defenses. The mentioned works are empirical defenses where the defender focuses on minimizing the effect of adversarial data and evaluates the performance empirically. Given that the adversary knows the details of the defense, there is always a possibility that the adversary can circumvent the defense and degrade the performance further. Our work, on the other hand, is a method that is provably robust against any attack under our specified threat models. Even if the adversary uses their perturbation budget perfectly, the robust model's performance is still at worst the reported ones in the paper. Therefore, it is not possible to compare these works with ours directly, as the performance we report is the worst case performance under the most effective attacks possible, while the evaluation in the mentioned works comes without a robustness guarantee.
>
> Secondly, the mentioned works mainly target evasion attacks, where an adversary tries to degrade model performance at test time by manipulating test samples. Although there is a brief discussion in GOOD-AT[1] addressing poisoning attacks, where the training data is adversarially manipulated, their defense is largely based on Li et al. 2023[2], which does not involve self-training. In GOOD-AT, self-training is introduced with an MLP only for test nodes, but this can still be evaded by a targeted adversary, whereas in our work, the performance is guaranteed by the partitioning scheme. In GPR-GAE[3], edges are generated in training time to train a pruning module in order to minimize the effect of edge perturbations at test time, while our edge generation is designed to enhance the individual classifiers directly by alleviating the sparsity of partitions. Although our works share the edge prediction element, they are used differently in different contexts.
>
> Overall, we propose a method for certifiable robustness that, in contrast to the mentioned empirical defenses, provably guarantees robustness. This discussion is now also included in our paper (line 75 on Page 2) for a more comprehensive view in the introduction.
>
> [1] Li et al., Boosting the adversarial robustness of graph neural networks: An ood perspective, ICLR 2024
>
> [2] Li et al., Revisiting graph adversarial attack and defense from a data distribution perspective, ICLR 2023
>
> [3] Woohyun Lee and Hogun Park, Self-supervised Adversarial Purification for Graph Neural Networks, ICML 2025
>
> **Weakness 2 & Question 2 (More challenging attacks)**
>
> We would like to clarify that our reported certified accuracy are provable lower bounds to any kind of attack and thus, assumes the strongest threat model possible including any imaginable attack, including white-box optimization-based attacks. We also want to clarify that the mentioned attacks PR-BCD & LR-BCD are only applicable to evasion attacks (attacking the test data), while our certificate concerns poisoning attacks (attacking the training data). However, as we do not propose an empirical defense which needs to be empirically tested against several strong attack strategies, but mathematically prove that our reported certified accuracies hold for any possible attack, stronger attacks cannot change our provable guaranteed lower bound on the accuracy, as they are already included.

---

> ### Author Response · Authors · 2025-11-24
> **Response to reviewer qsA6 (Part 2)**
>
> **Weakness 3 & Question 3 (Clarification that non-i.i.d. doesn't matter)**
>
> We thank the reviewers for raising this question. We bypass the i.i.d. assumption, by recognizing that to achieve the affect that poisoning one data point only affects one partition does not depend on assuming i.i.d. data, but on choosing an appropriate hash function that ensures that each poisoned object, like an edge or node label, is partitioned into exactly one partition and can thus, only affect this partition. Crucial to achieve this is to choose a hash function that does not depend on the poisoned parts of the non-i.i.d. structured data, but only on the parts which we know are not affected by the poisoning threat model (e.g., the node features).
>
> We would like restate Lines 139-140: "deriving a guarantee **like** Theorem 1 does not depend on the i.i.d. nature of the dataset". We mean here that a theorem similar to Theorem 1 from DPA can be derived, by adapting the proof to work with more complex hash functions operating on non-i.i.d. structured data such as graphs, resulting in our Theorem 2 in the paper. We clarify the dependence on the hash function in Lines 141-145 in the manuscript, which we now updated to make its effect to bypass the i.i.d. assumption more explicit. We are happy to provide further details and clarifications on this point, and also like to refer to the proof of Theorem 2 in Appendix C.1 (Page 19) to see that the i.i.d. assumption is not necessary to derive a guarantee similar to Theorem 1.
>
> **Weakness 4. (Heterophilic graphs)**
>
> Indeed, our technique to generate pseudo-edges depends on the homophily assumption in graphs and thus, won't be immediately effective when applied to heterophilous graphs. We now acknowledge and discuss this limitation and the general application to heterophilic graphs in our Discussion section in the main draft in Section 7 (Page 9).
>
> We want to note that our general certification scheme could be applied to heterophilic graphs by appropriately exchanging the individual steps (e.g., the link prediction module) with an equivalent method designed for heterophilic graphs, which we think is an interesting direction for future study.
>
> **Weakness 5. (Presentation)**
>
> We thank the reviewer for pointing this out. For the variable $r$ in Theorem 1, we were referring to the attack size (radius) introduced by the perturbation model. We have clarified this in the Theorem (Page 3). We thank the reviewer again for helping us to improve the clarity of our presentation. We have also increased the font size in Fig. 1 (Page 2) for better readability.

---

### Official Review · Reviewer_913P · 2025-11-05

**Soundness:** 2
**Presentation:** 3
**Contribution:** 2
**Rating:** 4
**Confidence:** 3

**Summary:**

This paper addresses certified robustness of GNNs against training-time poisoning attacks. The authors identify that existing partition-and-aggregate approaches used in image domain fail on graph data due to label and structure sparsity. They propose self training graph partition aggregation (ST-GPA), which augments partitions with pseudo-labels and synthetic edges through semi-supervised learning to enable effective certification.

**Strengths:**

1. Certified robustness against poisoning for GNNs is understudied compared to test-time attacks.
2. Using semi-supervised learning (label propagation, self-training, link prediction) to enrich sparse partitions seems to work.

**Weaknesses:**

1. The entire defense hinges on the self-training (label propagation, label self training, and link prediction self training) stage. However, these SSL methods are themselves run on the poisoned training data before certification. Could an attacker poison labels/edges in such a way that the label propagation step generates incorrect pseudo-labels that reinforce the attack?
2. The experiments do not address the scalability of this approach to larger datasets like OGB. How does the proposed idea scale with the number of nodes, edges, and partitions k?
3. How does the proposed work apply to non-homophilic graphs datasets? All the experiments are for homophilic graphs.
4. Some grammatical issues are there. Please work on addressing them.

**Questions:**

Please refer to the weaknesses.

---

> ### Author Response · Authors · 2025-11-24
> **Response to reviewer 913P**
>
> We sincerely appreciate the reviewers’ careful reading and helpful remarks. In the following, we respond to each point to address the weaknesses and questions noted.
>
> **Weakness 1. (Label Propagation is affected by poisoning)**
>
> Our robustness guarantees provide provable lower bounds on the accuracy given a certain perturbation budget. This means that *any* poisoning attack within that budget provably cannot reduce the accuracy more than what we report as certified accuracy. Any attack one could imagine, also on the label propagation step, is already included in our reported results, as they provably hold for a worst-case attack. Thus, under the given attack budget, an attacker is not able to change our reported certified accuracies by particularly targeting the label propagation step.
>
> **Weakness 2 (Scalability to larger graphs)**
>
> In response to the reviewer's request, we now provide a scalability analysis w.r.t. the number of nodes, edges and partitions in Appendix F (Page 24), as well as results on larger datasets like ogbn-arxiv in Fig. 12 of Appendix B.2 (Page 17). Runtime scales linearly in the number of partitions $k$, as the individual partitions are disjoint after the partitioning and can be trained identically and individually. Our label semi-supervised training scheme scales very well with the size of the graph (near linear), while our link prediction takes more time as the number of edges goes up. In larger graphs, the time for link prediction becomes predominant, compared to smaller graphs, where link prediction takes only negligible time. Further details are discussed in Appendix F (Page 24) and the limitation on link prediction is discussed in Section 7 (Page 9).
>
> **Weakness 3. (Heterophilic graphs)**
>
> Our certification scheme allows applications to heterophilic graphs if one would appropriately exchange the label propagation, self-training, or link-prediction steps with equivalent methods designed for heterophilic graphs. So far, the label propagation, self-training, and link prediction methods we leverage inherently utilize the homophily assumption in graphs and thus, aren't immediately applicable to heterophilic graphs. Since adapting these semi-supervised learning methods to heterophilic graphs are individual research directions on their own, we think this to be beyond the scope of this paper.
>
> However, our insights are still immediately relevant for certifiable robustness on heterophilic graphs, as a simple partitioning scheme still results in too sparse graphs for GNNs and a tailored semi-supervised learning scheme has to be used to obtain practical certified accuracies from GNNs. Thus, we think that exploring heterophilic graphs is an interesting direction for future research, and we now discuss the application to heterophilic graphs in our Discussion section in the main draft in Section 7 (Page 9).
>
> **Weakness 4. (Grammar)**
>
> We thank the reviewer for this comment. In response, we fixed several grammar mistakes and polished the language. We will further consider this for the camera-ready version again. If there are particular mistakes that haven't been fixed yet, please let us know!

---

### Author Response · Authors · 2025-11-24
**General Comment**

We would like to sincerely thank all the reviewers for their careful reading of our manuscript and their valuable, constructive feedback, recognizing (1) the clear identification of limitations in naive partitioning methods (Reviewer 913P, Reviewer qsA6, Reviewer m8sV), (2) the strong performance of our approach, and (3) well-organized presentation of our robustness certificates (Reviewer Zr4m). In response to their comments, we provide several new experiment results to strenghen the empirical evidence. All changes in the text are marked as blue text.

Concretely, we provide the following additional results:
- Certified accuracies on Wiki-CS and ogbn-arxiv against label flipping in Fig. 12 (Page 17);
- Analysis of scalability with respect to graph size and number of partitions in Appendix F (Page 24);
- Addition and comparison with a new baseline [1] in Appendix E (Page 23);
- Clean accuracies of vanilla GNNs for reference in Fig. 4 (Page 7);
- MLP performance for reference in Table 2 in Appendix A (Page 13);
- Degradation of ensemble classifiers' clean accuracies without our proposed semi-supervised learning methods in Appendix B.1 (Page 15).

And we make the following improvements in the text:
- Discussion of emprical defenses using self-training in Section 1 (Page 2);
- Clarification of the variable $r$ in Theorem 1 (Page 3);
- Clarification in Section 6 (Page 7) that our improved clean accuracy is compared to a partitioning-based certificate without proposed semi-supervised learning;
- Discussion on the application of our method to heterophilic or large graphs in Section 7 (Page 9);
- Discussion on related defense against label flipping in Section 8 (Page 10).

We believe these updates address the reviewers’ concerns and strengthen the paper. We are happy to clarify any remaining concerns or engage in further discussion.

[1] Li et al. "Deterministic certification of graph neural networks against graph poisoning attacks with arbitrary perturbations", CVPR 2025

---

### Author Response · Authors · 2025-12-02
**Final Response**

We would like to thank the reviewers again for their time and effort in reviewing our paper and for acknowledging the effectiveness and importance of our proposed approach for certifiable robustness of GNNs against poisoning attacks.

During the rebuttal period, we have improved our paper with several additional results in response to the Reviewers' comments, including (1) evaluations on larger datasets like Wiki-CS and ogbn-arxiv, (2) reference baseline performances like MLP accuracies and ensemble clean accuracies, and (3) a scalability analysis. Our additional experiments further support our main contribution. We have also addressed other concerns of the Reviewers regarding adaptive attacks and relationships to other self-training applications and self-training defenses (we clarified this in our responses and also added corresponding clarifications in our improved manuscript).

Overall, we believe that our contributions and insights are highly valuable for the research community. We provide a promising approach for machine learning on sparse, graph-structured data that offers more accurate and certifiably robust predictions under poisoning attacks, opening up promising directions for future research on robust machine learning on graphs and beyond.

---

### Meta-Review · Area_Chair_Ci7X · 2025-12-24

**Summary:**

This paper addresses the problem of robustness in Graph Neural Networks under both label and structural poisoning attacks. It clearly articulates the unique challenges of extending certification-based robustness techniques from images to graph-structured data. To mitigate these issues, the authors propose a self-training–driven ensemble approach that aggregates multiple classifiers to produce more reliable predictions.

Several reviewers raised concerns about the motivation, writing details/grammar errors, baseline settings, experimental data/scenario settings of the paper. Although the authors have provided the relative response, its credibility remains limited.   The manuscript would benefit from a substantial reduction in grammatical errors in future revisions, as well as a more thorough discussion of the method’s limitations across a broader range of scenarios.   In addition, the value and contribution of the proposed approach under its restricted applicability should be more clearly discussed.   Considering the negative feedback from all the reviewers and the absence of clear indications that these concerns can be adequately addressed, I decide to reject this paper.

**Reviewer Concerns:**

Regarding the reviewers’ concerns about the clarity of the paper’s motivation and method design, the authors have provided strong responses, effectively demonstrating their contributions through additional experiments and detailed explanations.
With respect to the inherent limitations of the proposed approach—such as the scalability challenges of link prediction on large-scale graphs and its incompatibility with non-homophilic settings—the authors have offered comprehensive clarifications.  Nevertheless, these fundamental limitations may still constrain the reviewers’ overall assessment of the work.
Finally, while the authors have corrected typographical and grammatical errors in the theoretical and textual presentation, such issues should not have occurred in the first place, and addressing them is unlikely to influence the final evaluation of the paper.

**Reviewer Scores:**

For Reviewer 913P, I don't think the reviewers will increase the score due to the inherent limitations of some methods.
For Reviewer qsA6, the author's response may have limited credibility and cannot prompt the reviewers to turn to a positive evaluation. Therefore, I think it might still keep the score unchanged.
For Reviewer m8sV, through multiple rounds of responses, I believe the author can fully address the reviewers' issues. Therefore, the score might be changed to 6 points.
For Reviewer Zr4m, the reviewer expressed a strong negative attitude towards the article as a whole. Although the author's response might raise the score to 4, I still think it remains a negative evaluation.
Overall, I think the final score still remains around 4 points. Therefore, I have decided to reject this article in view of the overall negative opinions.

---

### Decision · Program_Chairs · 2026-01-26

Reject